# Cell-to-cell expression dispersion of B-cell surface proteins is linked to genetic variants in humans

Gérard Triqueneaux[1,5], Claire Burny[1,3,5], Orsolya Symmons [1,4,5], Stéphane Janczarski[1], Henri Gruffat[2] & Gaël Yvert [1✉]

Variability in gene expression across a population of homogeneous cells is known to influence various biological processes. In model organisms, natural genetic variants were found that modify expression dispersion (variability at a fixed mean) but very few studies have detected such effects in humans. Here, we analyzed single-cell expression of four proteins (CD23, CD55, CD63 and CD86) across cell lines derived from individuals of the Yoruba population. Using data from over 30 million cells, we found substantial inter-individual variation of dispersion. We demonstrate, via de novo cell line generation and subcloning experiments, that this variation exceeds the variation associated with cellular immortalization. We detected a genetic association between the expression dispersion of CD63 and the *rs971* SNP. Our results show that human DNA variants can have inherently-probabilistic effects on gene expression. Such subtle genetic effects may participate to phenotypic variation and disease outcome.

[1] Laboratory of Biology and Modeling of the Cell, Univ Lyon, Ecole Normale Superieure de Lyon, CNRS UMR5239, Universite Claude Bernard Lyon 1, 69007 Lyon, France. [2] CIRI-Centre International de Recherche en Infectiologie, Universite Claude Bernard Lyon 1, Univ Lyon, Inserm U1111, CNRS UMR5308, Ecole Normale Superieure de Lyon, 69007 Lyon, France. [3] Present address: Institut für Populationsgenetik, Vienna Graduate School of Population Genetics, Vetmeduni Vienna, Vienna, Austria. [4] Present address: Max Planck Institute for Biology of Ageing, Cologne 50931, Germany. [5] These authors contributed equally: Gérard Triqueneaux, Claire Burny, Orsolya Symmons. ✉email: Gael.Yvert@ens-lyon.fr

Stochasticity in molecular processes can cause biological differences between genetically identical cells, and this heterogeneity can affect development, adaptation, as well as disease emergence or relapse[1–4]. Because of these implications, it is important to understand if there is a genetic basis for cell-to-cell heterogeneity. For example, in the case of a drug treatment, if a genetic variant increased heterogeneity, then a larger sub-population of cells could be in a refractory state at the time of treatment, thereby compromising the desired clinical effects (see review[4]). Using model organisms and single-cell measurements, we and others have shown that genetic variants can indeed tune cell–cell heterogeneity[5–11]. Consequently, we previously introduced the concept of single-cell Probabilistic Trait Loci (scPTL)[2]. This term is analogous to Quantitative Trait Loci (QTLs) but defines genetic variants that control single-cell traits in ways that can be more subtle than changing the trait's mean. Such subtle effects can be considered as 'non-deterministic'—or 'inherently probabilistic'—because they change the probability that one cell displays a given trait value at a given time without necessarily affecting the average of the cell population. Note that, if the trait results from a stochastic process, a deterministic effect on a kinetic parameter of the process can have inherently probabilistic effects at the cell level. For example, a change of transcriptional burst frequency and size may modify gene expression probabilities without necessarily affecting mean expression[12]. As discussed earlier[2], scPTL may contribute to disease predisposition and, given the above-mentioned link between heterogeneity and proper cell eradication by drugs[4], they may also contribute to the outcome of specific treatments. It is therefore essential to investigate the existence and properties of scPTLs in humans.

In this study, we searched for human scPTL that affect cell-to-cell variability of gene expression traits. Because differences in mean expression levels are usually accompanied by differences in variability[5,13,14], we took special care to distinguish genetic loci that modulate variability per se from those that affect mean expression, and thereby variability. In other words, we were interested in expression 'dispersion' (variability after accounting for the mean). In model organisms, the distinction between dispersion and variability can be made by applying dedicated genetic crosses[8,15]. In humans, however, covariation of mean and variability cannot be decoupled experimentally. Even if single-cell data can provide fine-scale descriptions of the genetic control of gene expression[16], loci affecting variability are sometimes identified simply because of their effect on mean expression[17].

The existence of human scPTL of gene expression variability per se is supported by the observations of Lu et al.[18] who identified loci that affect gene expression variability in primary T cells of specific subtypes. These loci affected variability in ways that were not explained by a modulation of the mean. This work was therefore essential in establishing that scPTL with probabilistic effects exist in humans under physiological conditions. However, primary T cells comprise a complex mixture of cells, and could therefore possibly include various sources of heterogeneities (see Discussion). For this reason, we sought here to investigate variability among cells of the same type and grown in identical conditions, using cultured cell lines derived from different donors. Although less representative of the variability that is present in vivo, this approach offers better control of the cells' subtypes and the environment.

We utilized lymphoblastoid cell lines (LCLs) from the 1000 Genome Project Consortium[19]. These cell lines have been extensively genotyped and, being derived from B cells, they express surface proteins that can easily be quantified on single cells via immunostaining and flow cytometry. Using this resource, together with freshly-generated LCLs and subcloning experiments, we found that the level of expression dispersion of four cell-surface proteins (the low-affinity receptor for IgE CD23, the Decay-Accelerating Factor CD55, the tetraspanin CD63 and the co-stimulator CD86) differs between individuals. In addition, we detected an association between the expression dispersion of CD63 and the genotype at a SNP located in cis. Our results illustrate that naturally occurring genetic variants can change gene expression parameters other than the mean. Such variants are missed by classical studies based on bulk measurements although they may have important phenotypic consequences.

## Results

**Terminology**. This study describes differences in gene expression both between cells and between individuals. It is therefore important to clarify the meaning of several terms that are used hereafter. Expression variability will refer to differences in expression level of a protein between cells of the same genotype, same cell type and subtype, and which are extracted simultaneously from the same environment; given that expression variability often co-varies with mean expression, we also use the term expression dispersion of Sarkar et al.[17] to refer to the amount of expression variability that is not explained by the mean; an individual will refer to a human person; variation will refer to differences of a given scalar value, for example a summary statistics of single-cell values, between individuals having different genotypes; cell line will refer to a population of immortalized cells of the same cell-type that can be propagated in vitro and which derives from a single donor individual; clone will refer to a cell line deriving from a single primary cell extracted from an individual; sample will refer to a population of cells that were cultured together in a single well and collected in a single tube for investigation.

**Quantifying expression dispersion among lymphoblastoid cells**. Comparing cell-to-cell expression dispersion across individuals of a cohort presents several difficulties. First, it requires acquisitions on single cells that are all of the same type (or subtype) and share a common environment. Second, this cell-type and environmental context must be the same when analyzing all individuals of the cohort. Third, as for any trait, inter-individual variation of expression dispersion can only be interpreted if intra-individual variation is also estimated. Finally, a large number of cells must be analyzed to obtain robust estimates of dispersion. Lymphoblastoid cell lines represent a powerful resource to face these challenges. They have been invaluable for characterizing human genetic variation, and they were widely used to map quantitative trait loci (QTLs) of various molecular and cellular traits, including gene expression[20]. LCLs are derived from B-lymphocytes through immortalization by Epstein–Barr virus (EBV) infection. The advantage of lymphoblastoid cells is that they express many B cell-specific cell-surface proteins, for which monoclonal antibodies have been developed and that can therefore easily be quantified on single cells. By using fluorescent-conjugated antibodies and flow-cytometry, as routinely done in immunological studies, it is possible to quantify the expression of an antigen in tens of thousands of cells in minutes. This can provide robust estimates of expression dispersion. Moreover, since many of these cell lines have been extensively genotyped, it is possible to search for an association between dispersion and genotype by applying linkage tests if differences in dispersion are observed between cell lines from different donors.

To make full use of this experimentally tractable system we selected cell lines from the Yoruba population in Nigeria that had previously been genotyped as part of the HapMap and 1000 Genomes Project[19]. We selected Yoruba samples, since previous studies had shown them to have the largest genetic diversity,

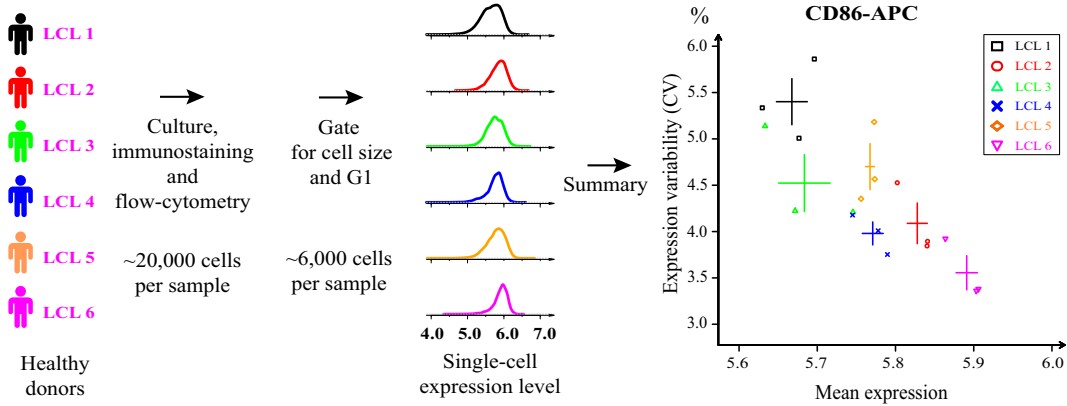

**Fig. 1 Experimental single-cell gene expression quantification in LCL lines.** Cell lines from different healthy donors were cultured, fixed, immunostained with a fluorescent antibody, and analyzed by flow cytometry. A computational pipeline (see methods) automatically gated cells with similar size and in the G1 phase of the cell cycle, yielding distributions of expression values (log of fluorescence intensity) of the antigen protein of interest (here CD86). Summary statistics of the expression distributions were computed, such as the mean and coefficient of variation (CV = sd/mean). The experiment (culture, staining and acquisitions) was repeated several times independently to estimate intra-line variability (error bars: s.e.m., here $n = 3$ samples). Differences in CV between lines displaying similar mean levels of expression reflects different levels of cell-to-cell variability in expression (visible here for LCL4 versus LCL5). Icons of persons were modified from an icon created by Muhammad Haq and provided by https://www.freeicons.io under the Creative Commons (Attribution 3.0 Unported) licence.

which is favorable for genetic mapping. For all experiments, cells were cultured, fixed, immunolabelled with a fluorescent antibody directed against a cell-surface protein of interest, stained with DAPI and analyzed by flow-cytometry. We developed and applied a dedicated analysis pipeline to automatically gate cells of similar size and in the G1 phase of the cell-cycle, yielding distributions of single-cell protein expression levels (Fig. 1). The experiment, from culture to acquisition, was repeated several times so that we could compare the mean and coefficient of variation (CV) of these distributions within and between cell lines.

**Survey of 18 surface proteins in 6 unrelated individuals.** We started with a pilot survey using 6 cell lines from unrelated healthy Yoruba donors. We searched the literature and selected a set of 19 proteins meeting the following criteria: (i) evidence of expression in LCLs, (ii) availability of a validated monoclonal antibody suitable for fluorescent immunostaining and, (iii) these proteins participate to various biological functions of B cells, including disease-related processes. For one protein, ROR1, the fluorescent signal was not above the background signal obtained from unstained cells. We therefore proceeded with the remaining 18 proteins only, for which we applied our labeling and analysis protocol and analyzed each cell line in triplicate (independent cultures and staining). We computed the mean and CV of protein expression of the corresponding population of cells (Fig. 2a) and we examined if one or more of the proteins displayed different CV across individuals. Four proteins (CD2, CD9, CD37, CD79b) were not or very poorly detected in all 6 cell lines. Four proteins (CD40, CD46, CD59, CD80) were detected but displayed no marked differences in either expression mean or variability between cell lines. One protein (CD20) displayed mild variation in mean but not in variability. One protein (CD53) showed differences in variability between cell lines but it was weakly detected in all samples. Four proteins (CD5, CD22, CD38) displayed reliable differences of variability between cell lines, but this variation was fully correlated with variation of the mean (lower CV at higher mean). The remaining 5 proteins (CD19, CD23, CD55, CD63, CD86) were reliably detected, displayed variation in both mean and CV of expression, and, interestingly, variation in CV was not completely explained by variation in the mean. For

example, the CV of CD23 expression decreased linearly with its mean expression across five cell lines but was lower in the sixth cell line (arrow on Fig. 2a); and for CD55, one cell line (orange color in Fig. 2) seemed to have a larger CV but a similar mean expression as compared to two other cell lines. Example distributions are shown in Fig. 2b. Thus, our survey identified five proteins that displayed different levels of expression dispersion among six Yoruba cell lines.

**Single-cell expression distributions display specific patterns of variation.** We decided to further characterize the single-cell expression properties of four of the five proteins: CD23, CD55, CD63, and CD86. All 4 proteins are involved in modulating the immune response, but have very different functions. CD23 is the low-affinity receptor for IgE but can also bind CD21, major histocompatibility complex class II proteins and integrins; it has several isoforms and it can be cleaved to release a functional soluble form[21]. CD55 is the Decay-Accelerating Factor, which protects cells from the complement system[22]. It has additional roles by transducing signals in T cells[23]. CD86 is a co-stimulator of CD28-mediated T cell activation[24]. CD63 is a tetraspanin protein with numerous regulatory roles[25], including exosome-based co-stimulation of T cells.

We cultured, immunostained and analyzed a total of 50 Yoruba cell lines, in 6 biological replicates for each antigen and cell line. Note that, to avoid any biases of signal amplifications or crosstalk between fluorescent channels, each sample was stained with only one, directly-labeled antibody. We processed the data as above to derive distributions of expression values of G1-gated cells. We first plotted, for all samples, the CV (variability) of these distributions as a function of the mean (Fig. 3a). The four proteins clearly differed in their pattern of variation. CD23 was very poorly expressed in two cell lines and displayed wide variation of mean expression across the remaining 48 lines. Its CV of expression varied among moderately-expressing cell lines and decreased abruptly in cell lines having high mean expression. The spread of mean expression and CV was also large for CD55 and CD86, with an important distinction: for CD55, the CV decreased linearly with the mean, whereas for CD86, variation of CV and mean were independent (Fig. 3a). CD63 displayed the lowest variation in both mean and CV, although the CV was

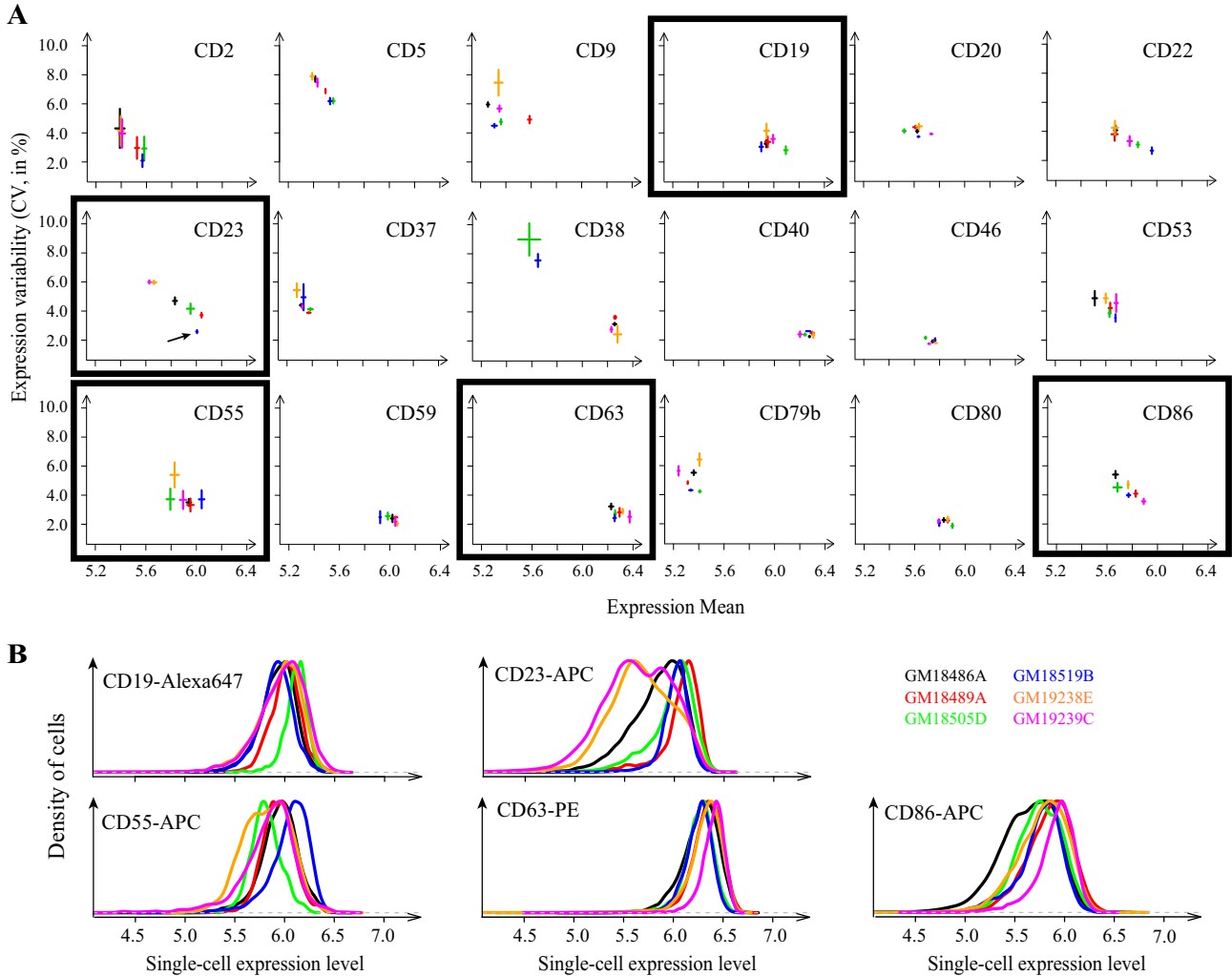

**Fig. 2 Single-cell expression of surface proteins in cell lines from unrelated individuals. a** Six cell lines from unrelated individuals (one per color) were immunostained for the indicated protein and analyzed by flow-cytometry to estimate CV and mean expression in cell populations. Each dot represents mean ± s.e.m. between biological replicates of the same cell line; $n >= 3$ except for CD38 where $n = 1$ (dot instead of bars) for one cell line and $n = 2$ for 4 cell lines. Arrow: the CV of CD23 was significantly different between GM18519B and GM18489A (t-test $P = 0.003$). **b** Distributions of single-cell expression values for 5 of the proteins shown in the boxed panels of **a**. Each distribution corresponds to one randomly-chosen sample of the indicated protein and cell line (same color as in **a**).

clearly elevated in three cell lines. We quantified expression dispersion of these three proteins by applying a locally-linear (lowess) regression to the CV versus mean dependency. An example of this regression is shown in Fig. 3b. We used the residuals of the model (noted CV|mean, or conditioned CV) as estimates of expression dispersion. Figure 3c shows the spread of variation among cell lines of expression dispersion values. This variation was larger than the variation observed among replicates of the same line. Of the four proteins, CD86 was the one displaying the largest line-to-line variation of expression dispersion. We verified that this variation was not an artifact of cell fixation: repeating the experiment with immunostaining on live cells produced very similar single-cell expression distributions as with fixed cells (Supplementary Fig. 1).

Two possible patterns of variation could take place: a global variation, where some individuals would display high dispersion for all four proteins; or a modular variation, where individuals could display elevated dispersion for some of the proteins but not others. To examine this distinction, we tested if the expression dispersion values of different proteins were correlated, i.e., whether individuals with elevated dispersion for one protein also

display high dispersion for another protein. We computed all pairwise correlations between dispersion values in the 50 LCLs (Fig. 4). We found positive correlations for three pairs of proteins (CD23/CD63, CD23/CD86, CD55/CD86), and clearly no correlation for one pair (CD55, CD63). Thus, covariation of dispersion exists for some of the proteins, but not for all. We conclude that the pattern of variation is modular.

**Co-existence of high-expressing and low-expressing CD23 cells.** We made the interesting observation that, in some but not all LCLs, the distribution of single-cell CD23 expression was bimodal. This was also true when we immunostained live cells (Supplementary Fig. 1); ruling out an artifact caused by cell fixation. In such cases of bimodality, the mean and CV of the distribution do not capture all properties of single-cell expression. We therefore sought to provide a more comprehensive description by fitting a mixture model to the observed data. The model consisted of two Gaussians that each described a subpopulation of cells. The model was fully defined by five parameters: the means ($\mu_1$, $\mu_2$) and standard deviations ($\sigma_1$, $\sigma_2$) of the Gaussian components, and the

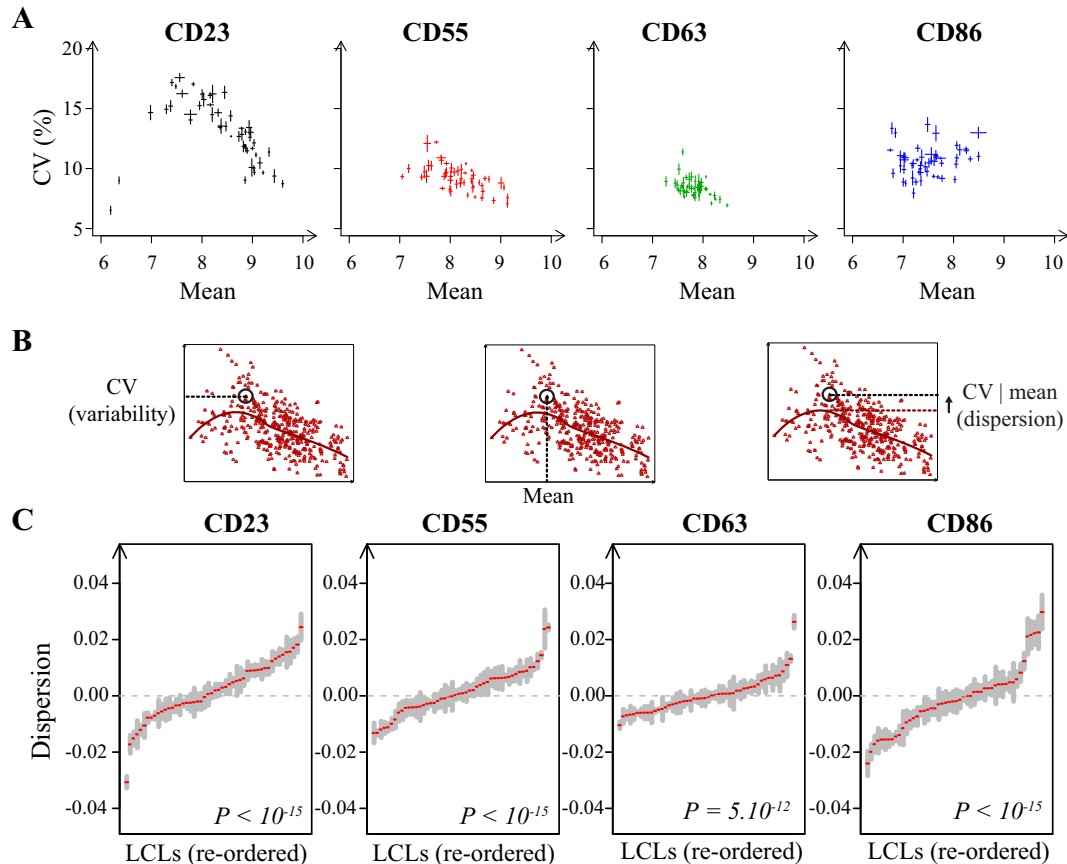

**Fig. 3 Patterns of variation in mean expression and CV across 50 individuals. a** For each sample (at least 6 per antibody and cell line), the CV and mean expression were computed. Each dot represents mean ± s.e.m. computed for each cell line and antibody combination. **b** Lowess regression applied to the data (here for CD55 expression). The model was fitted to the entire dataset and for each sample, residuals were extracted (CV|mean = difference between the observed CV and the CV predicted by the model given the observed mean expression). **c** Distributions of expression dispersion of the indicated proteins in 50 Yoruba cell lines. Each red tick corresponds to the average value of dispersion across biological replicates. Gray shaded bar = +/− s.e.m. ($n > = 6$). $P$: Kruskal–Wallis rank sum test.

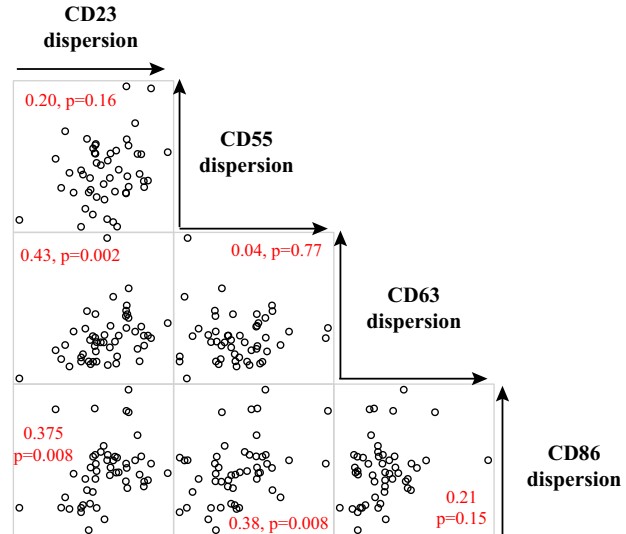

**Fig. 4 Correlation between expression dispersions of different proteins.** Each dot plot represents expression dispersion values (CV|mean) of two proteins (which names appear on the diagonal) across 50 Yoruba LCLs (one per dot). The Spearman rank correlation coefficient is indicated on each plot with the corresponding $p$-value.

proportion of cells belonging to the first component ($p_1$). By fitting model parameters to the data of each CD23-stained sample, we compared how these distributions differed between cell lines. Hierarchical clustering of the cell lines based on parameter values revealed three groups of lines (Fig. 5). Cluster 1 contained only three LCLs and was characterized by a majority of cells with low CD23 expression and a right tail of cells at higher CD23 levels. A second group containing half the LCLs had a mirrored pattern, with a bulk of cells at high expression and a left tail of low-expressing cells. For the remaining 22 LCLs, two sub-populations of cells defined by distinct modes of expression clearly co-existed. Thus, cell populations can differ not only in expression dispersion but also in expression bimodality, where two distinct states can be identified in some individuals but not in others. For CD23, which is a low-affinity receptor for IgE, bimodality implies that one cell type simultaneously generates highly-responsive and poorly-responsive cells to low levels of IgE. Our results suggest that this duality can be pronounced in some but not in all individuals.

**EBV-mediated immortalization does not explain inter-individual differences in expression dispersion or bimodality.** The use of LCLs has been criticized because immortalization by EBV modifies cellular regulations as compared to non-infected primary cells. For example, CD23 is known to be upregulated in EBV-activated cells[26]. It is possible that EBV modifies not only the mean expression of host-cell genes but also their expression

**A**  **B**

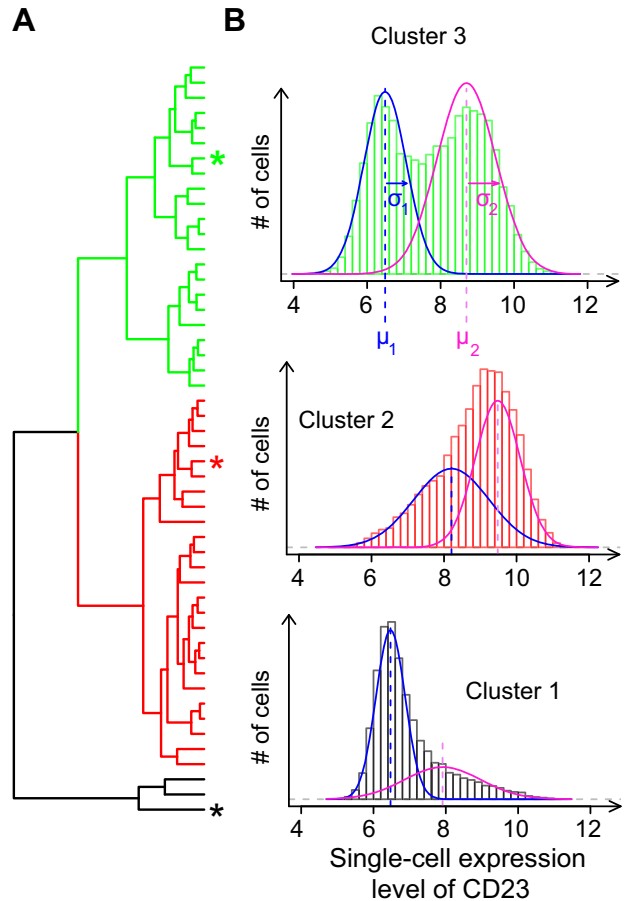

**Fig. 5 Variation in the bimodality of CD23 expression. a** Hierarchical clustering of LCLs based on GMM parameters. Clusters 1 to 3 contained 3, 25, and 22 LCLs, respectively. LCL labels are provided in Supplementary Fig. 5. **b** One sample was randomly chosen from each cluster to plot the observed data (histogram) and fitted model components (lines). These samples correspond to LCLs labeled as asterisk (*) in **a** (same color).

dispersion. In this case, independent immortalization events may generate cell lines with different levels of expression dispersion, regardless of original donor. It is therefore important to determine if the different levels of expression dispersion that we observed here could result from different outcomes of the immortalization process. To this end, we performed two complementary series of experiments.

We first reasoned that if differences observed between cell lines are primarily driven by inter-individual variation, then only few differences should be observed among cell lines originating from the same donor. Such variation between different cell lines from the same donor cannot be directly estimated from Yoruba LCLs because a single LCL is available per donor. We therefore generated additional cell lines from two unrelated and healthy donors. For practical reasons, these donors were not from the Yoruban cohort. Primary cells from blood samples were infected with EBV and twelve independent cell lines were established from each donor. We cultured and processed these lines to quantify the expression of CD23, CD55, CD63, and CD86 at the single-cell level as above. To study variation within and between individuals, we first plotted the mean, variability (CV) and dispersion (CV| mean) of expression for CD55, CD63, and CD86 (Fig. 6a, b). For all three proteins, the two donors differed in both mean and CV values. Very importantly, the spread of variation between lines of the same donor was much lower than between lines of different

donors. This was true not only for mean expression but also for CV and dispersion. In particular, all cell lines from one individual (donor 1 on Fig. 6) displayed markedly higher CD86 expression variability and dispersion than the cell lines from the other individual. Consistently, for all three proteins, variation in mean, CV and dispersion was larger among the Yoruba cell lines than among cell lines originating from a single donor. We analyzed CD23 single-cell expression distributions separately, taking into account bimodality. We fitted a Gaussian mixture model to the data of each sample and we analyzed variation of model parameters (Fig. 6c). For all five parameters of the model, variation among the Yoruba LCLs was clearly larger than the variation observed among lines of a single donor. These observations made on de novo LCLs show that there is a very high consistency of dispersion estimates from the same individual: although intra-individual line-to-line variation in expression dispersion exists, it is lower than inter-individual variation.

In a second series of experiments we tested if elevated dispersion or pronounced bimodality in some Yoruba cell lines could potentially emerge as a consequence of multiple EBV-mediated immortalization events, since this process can result in polyclonal cell lines if more than one primary cell is infected and expands in a given sample. If different clones contained in a cell line expressed a protein at slightly different levels, this inter-clone variability would be seen as cell-to-cell variability at the level of the whole cell line. We therefore sought to i) determine, for some of the Yoruba cell lines, if they were monoclonal or polyclonal and ii) quantify expression dispersion in the context of monoclonality.

To distinguish between monoclonality and polyclonality, we took advantage of the somatic VDJ rearrangements and editing that take place during B cell maturation. This process occurs exclusively in vivo, during a complex interplay between cell types within germinal centers, and generates the immunoglobulin specificity expressed by mature B cells. Cell lines of the 1000 Genome Project were established by infecting blood cells with EBV in culture dishes ex vivo. Thus, any specific VDJ sequence is a signature of one cell that matured in vivo before cell line generation, and clonality of LCLs can be assessed by their genetic homogeneity at VDJ junctions: if the population is monoclonal, only a single signature will be observed, while in a multiclonal population multiple signatures will be present. We adapted previously described protocols[27,28] to simultaneously amplify several fragments of the VDJ junctions (Fig. 7a). A secondary amplification was then used to tag amplicons with indexing primers informing on the sample of origin and allowing for multiplexed 150 bp paired-end Illumina sequencing.

To study the relationship between expression dispersion and clonality in LCLs, we subcloned six cell lines by limiting dilution (Fig. 7b, see methods), which yielded a total of 27 subclones. Using the method described above, we genotyped these—presumably monoclonal—subclones and their parental lines, in duplicates (see methods). To analyze this data, we developed a dedicated pipeline based on IgBlast[29] that produced unique peptide sequences of the variable CDR3 region generated by VDJ rearrangements and editing (see methods). Results of this genotyping are shown in Supplementary Table 1. We found that two of the parental cell lines (GM18505D and GM18486A) were monoclonal: their genotype was homogeneous and was found in all of their subclone progenies. We also found that three Yoruba cell lines (GM18519B, GM18489A, and GM19238E) were unambiguously polyclonal: their genotype was heterogeneous and genotypes of their progenies differed. Results on the sixth cell line (GM19239C) were inconclusive. Thus, the 1000 Genomes Project resource contains both monoclonal and polyclonal LCLs.

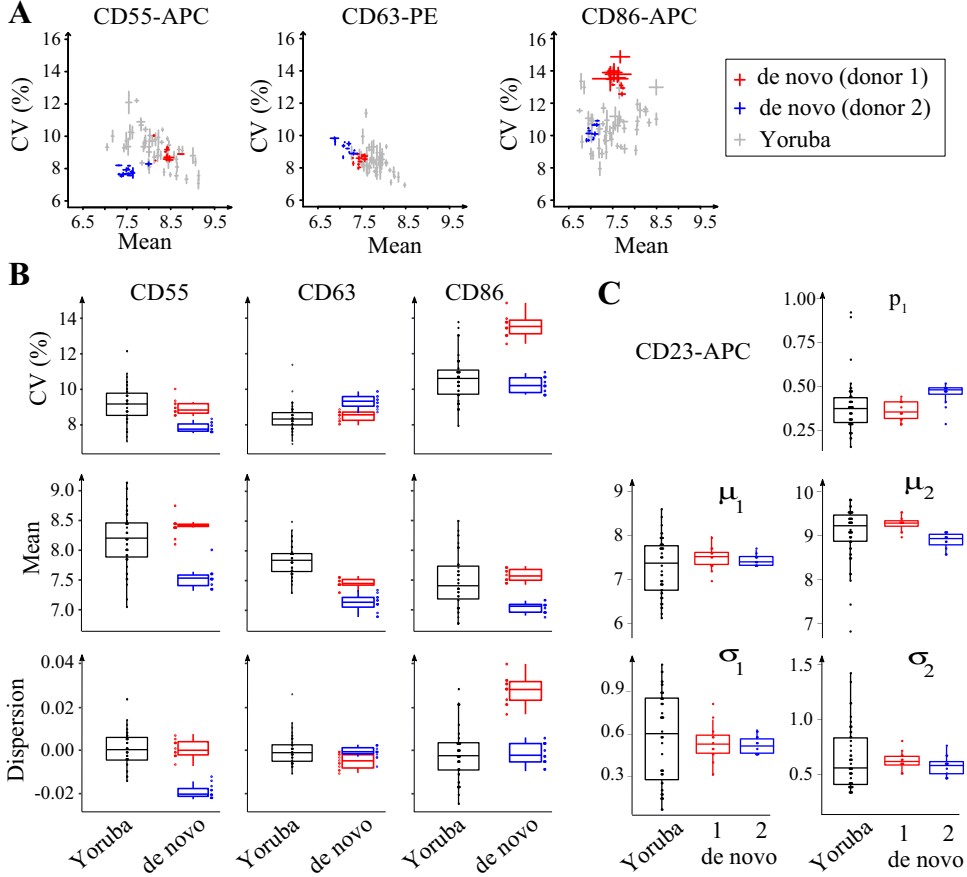

**Fig. 6 Intra-individual versus inter-individual variation of expression dispersion.** Multiple LCLs were generated de novo using blood samples from two unrelated donors (blue and red), and expression mean and variability was compared to those observed in Yoruba LCLs. **a** Dot plots of CV versus mean expression of the indicated proteins. Each dot represents one cell line, as mean ± s.e.m. (*n* = 2 independent cultures of each de novo LCL). **b** Boxplot of CV, mean, and dispersion. Dispersion values correspond to CV|mean residuals computed from a lowess regression fitted to all samples. **c** Boxplot of GMM parameter values fitted to distributions of single-cell CD23 expression (as in Fig. 5). Each dot represents one cell line. Values of replicate cultures were averaged.

This information may be important when interpreting variation in molecular traits among cell lines, especially for traits that are related to epigenetics. For example cell line GM19238, which we found to be polyclonal, was previously used in at least three studies of inter-individual variation of chromatin marks[30–32]. If these marks diverged between clones contained in the cell line, then the corresponding epigenomic trait values could be affected by the relative abundance of each clone in a sample.

Of the 27 subclones, 21 had a single CDR3 genotype and only 2 produced a clear mixture of genotypes (Supplementary Table 1). This validated the efficiency of the subcloning procedure. We therefore quantified expression dispersion in some of the subclones and their parental lines by immunostaining and flow cytometry as above.

Results for CD23 are shown in Fig. 7c. Of the four parental cell lines that we re-analyzed, three showed high expression dispersion. Two of these lines were polyclonal (8E and 9A) and one was monoclonal (5D). The remaining cell line displayed very little CD23 expression dispersion and was polyclonal (9B). Thus, polyclonality does not correlate with CD23 expression dispersion in these Yoruba cell lines. We also analyzed six subclones. Five showed CD23 expression distributions that were fully consistent with the distributions observed on their respective parental lines. Importantly, the pronounced dispersion of polyclonal lines 9A and 8E was reproduced in their monoclonal derivatives 3H10 and

3F7. This excludes the possibility that expression dispersion in these two Yoruba LCLs was due to their polyclonality. In addition, subclone 8E4 displayed marked bimodality of CD23 expression (co-existence of low and high-expressing cells) despite its confirmed clonality. Thus, bimodality persists after single-cell subcloning and, in this example, it does not result from polyclonality.

Results for CD55, CD63 and CD86 also excluded a systematic association between polyclonality and expression dispersion. For all three proteins, we could find a pair of subclones that differed from one another regarding variability in a way that was not explained by a difference in mean expression (arrows on Fig. 7d). For CD55, clone 3H10 had higher CV than 5D4 (*P* = 0.001, *t*-test) but similar mean (*P* = 0.7, *t*-test); for CD86, clone 3H10 had higher CV than 5D10 (*P* = 0.004, *t*-test) but similar mean (*P* = 0.6, *t*-test); for CD63 clone 8E4 had higher CV than 5D10 (*P* = 0.01, *t*-test) and its mean expression was not lower but even higher than 5D10 (*P* = 0.01, *t*-test). This demonstrates that LCLs can differ in expression dispersion of CD55, CD63 and CD86 despite being monoclonal.

Altogether, these observations on de novo and on monoclonal LCLs exclude the possibility that differences in expression dispersion between Yoruba LCLs simply result from variable outcomes of EBV-based immortalization. This motivated us to search for genetic factors that might underly these differences.

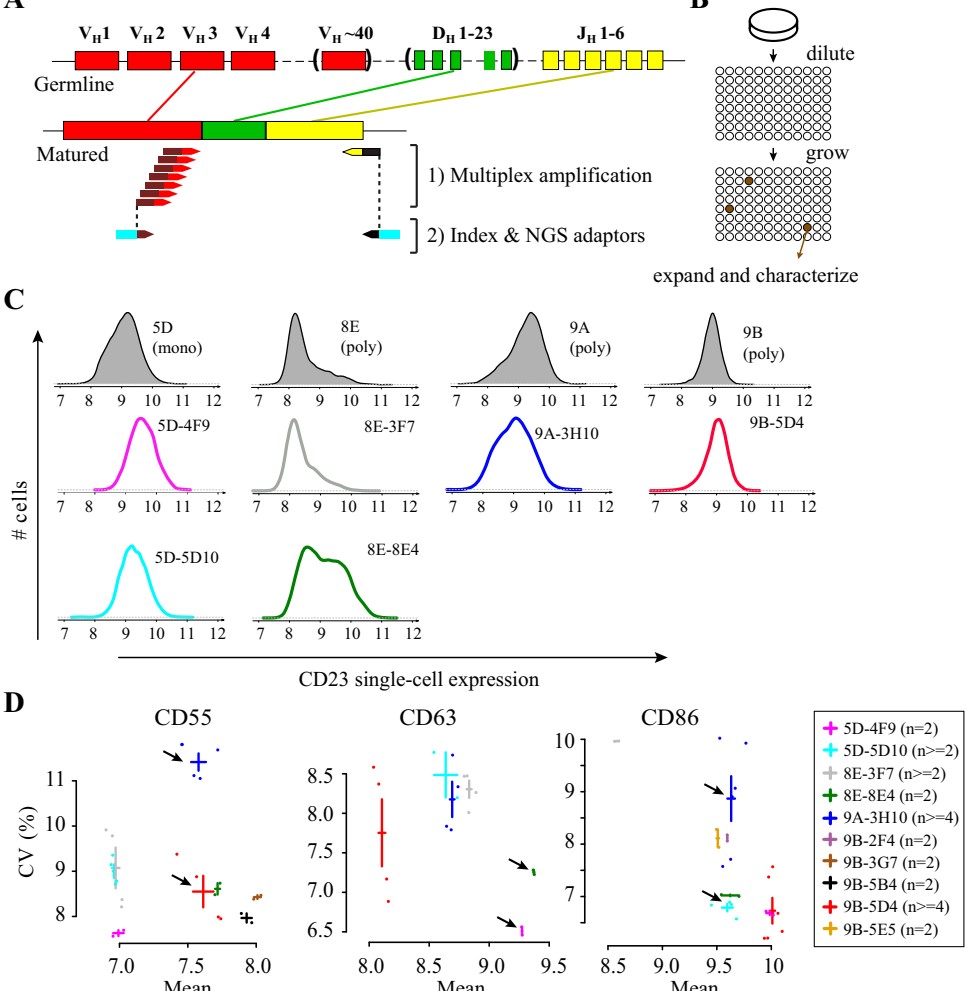

**Fig. 7 Subclones of LCLs also display variation in expression dispersion. a** Scheme of the clonality test applied to cell lines and subclones (see main Text and Supplementary Table 1 for results). **b** Subcloning procedure. **c** Single-cell expression distributions of CD23 in cell lines (shaded gray) and subclones (colored lines). 5D = GM18505D; 9A = GM18489A; 8E = GM19238E; 9B = GM18519B. **d** Expression mean and CV of three proteins in ten monoclonal subclones. Crosses: mean +/− s.e.m. *n*, number of independent samples. Arrows point to examples of subclones with statistically significant differences in dispersion (see text). In **c-d**, subclones nomenclature indicates their origin (e.g., 5D-4F9 derived from 5D).

**Genetic mapping of expression variability and dispersion**. We searched for association (genetic linkage) between DNA variants and gene expression dispersion using the available genotypes of the Yoruba LCLs. Of the fifty cell lines analyzed here, forty had been genotyped with phased and high-coverage data, eight were at an earlier stage with unphased data, and two were not characterized. We therefore applied linkage using either a high-quality map of SNPs covering 40 individuals, or a lower-density map covering 48 individuals. In each case, we searched for association between SNPs located in the vicinity of the gene of interest (+/−2 Mb from TSS) and expression traits: mean, variability, dispersion and, in the case of CD23, five fitted parameters describing bimodal distributions. We considered an association to be significant if its family-wise error rate was lower than 5% and we estimated, for each trait, the False Discovery Rate (FDR) corresponding to the retained associations.

Results are summarized in Table 1. We found no association for mean expression levels. This was consistent with an earlier report describing genetic regulations of mean mRNA and protein levels in 72 Yoruba LCLs, and where associations were found for two of the four genes studied here (CD23 and CD55) but only at the mRNA level and not at the protein level[33]. However, we did

find associations for expression variability and dispersion. For CD23, dispersion was associated with 8 clustered SNPs but this association was marginally statistically significant and supported by 4 individuals displaying reduced dispersion as compared to others (two individuals had barely-detectable expression, Supplementary Fig. 2a). For CD86, we found an association for both variability and dispersion using the 40 densely-genotyped individuals. This association corresponded to four heterozygous individuals displaying high dispersion. Notably, two individuals also displayed high dispersion but were not covered by the genetic map. To add them in the analysis, we genotyped them at the associated SNPs by PCR and sequencing. This revealed that they were homozygous, therefore eliminating the genetic association (Supplementary Fig. 2b). Finally, we found an association between CD63 variability and SNP *rs971* (Fig. 8a). This linkage was supported by both homozygous and heterozygous individuals, with one homozygous individual displaying high expression variability. Importantly, association was not accompanied by mean effect, and the genotypic groups also differed in expression dispersion (Fig. 8b). Note that our observations do not fully demonstrate the effect of *rs971* on CD63 dispersion because i) the genetic association needs to be replicated using another sample of

**Table 1 Results of genetic association tests.**

| Protein | Trait | #LCLs | Chromosome | Position (i) | SNPs | p-value (ii) | FDR (iii) | Conclusion |
|---|---|---|---|---|---|---|---|---|
| CD23 | CV\|mean | 40 | chr19 | 6246936 to 6277226 | rs75228027, rs191354398, rs16785610, rs115321934, rs116565827, rs201284176, rs114426250, rs115581057 | 0.045 | 0.014 | Marginal association (ii) (Supplementary Fig. 2A) |
| CD86 | CV | 40 | chr3 | 123464070 to 123492460 | rs16471535, rs145139961, rs150277702, rs144503460, rs16353991 | 0.021 | 0.02 | Not validated with 2 more individuals (Supplementary Fig. 2B) |
| | CV\|mean | 40 | chr3 | 123464070 to 123492460 | rs16471535, rs145139961, rs150277702, rs144503460, rs16353991 | 0.009 | 0.007 | |
| CD63 | CV | 48 | chr12 | 54575458 | rs971 | 0.014 | 0.006 | QTL of variability (Fig. 8) |

(i) These SNP positions were retrieved from genotype data and may therefore differ from more recent physical maps.
(ii) p-value was corrected for the number of SNPs tested, but not for the number of traits investigated (8 for CD23).
(iii) Benjamini–Hochberg procedure implemented in PLINK.

individuals and ii) the mechanism by which *rs971* affects CD63 expression dispersion remains to be found. The SNP resides ~1.5 Mb away from CD63, in the 3'UTR of SMUG1, a gene involved in base excision DNA repair (Fig. 8c). We inspected annotated positions of enhancers and transcription factor binding sites and found none overlapping *rs971*. Interestingly, according to dbSNP (www.ncbi.nlm.nih.gov/snp/) the *rs971* allele associated with high variability is not restricted to Yoruba but is present in all described human populations, with a minor allele frequency of at least 19%.

## Discussion

By quantifying a handful of cell-surface proteins in millions of individual lymphoblastoid cells, we found that the level of cell-to-cell expression variability can differ between healthy humans even within a single cell subtype and under controlled conditions. For proteins CD23, CD55, CD63, and CD86, variation in mean expression did not fully explain the differences observed on variability levels, demonstrating variation of expression dispersion among humans. Expression of CD23 was bimodal for some individuals (co-existence of two subpopulation of cells with different mean expression) but not for other individuals. Analysis of de novo-generated cell lines and subcloning experiments excluded the possibility that expression dispersion and bimodality systematically resulted from variable outcomes of the immortalization process. Consistently, we found a *cis*-acting SNP linked to CD63 expression variability independently of the mean. We note that, for a full demonstration of this genetic association, this detection still needs to be replicated on another sample of individuals.

Our results complement the study of Lu et al.[18] who analyzed single-cell expression of 14 proteins in various primary T cell subtypes from unrelated donors. Their experimental design and ours have several features in common: antibody staining and flow-cytometry, a focus on dispersion rather than variability—although the term used by Lu et al. was cell–cell expression variation (CEV)—and coverage of a cohort of healthy individuals allowing genetic linkage analysis. The two studies also have important differences. Lu et al. used primary T cells while we used lymphoblastoid cell lines; they quantified all proteins simultaneously by multicolor staining and compensation while we used single labeling; they interrogated ~800 SNPs that had previously been linked to immunological traits and that were located away from the gene for which expression was considered (*trans*-acting SNPs), while we scanned thousands of SNPs located at the gene locus (*cis*-acting SNPs); and their cohort included individuals from different origins. These specificities have implications on the interpretation of results. First, cells were not only from different types (T versus B) but were also extracted from a totally different context (primary versus cultured lines). In the case of Lu et al., the data from primary T cells directly reflects in vivo cell states and their work demonstrated that both genotype and age affect expression dispersion in a physiologically-relevant context. However, the nature of this dispersion is not easy to define in this case. Primary cells belong to a complex classification of subtypes and the definition of cell-cell variability heavily relies on how each subtype is defined. Lu et al. provided detailed reports by carefully stratifying the data (manual gating). Nonetheless, if a subtype is in fact structured in two or more subpopulations of cells, the variability that is measured can then include both stochastic variability within each subpopulation and differences in mean expression between subpopulations. In contrast, culturing cell lines ex-vivo avoids this difficulty because it generates cells that directly come from a single subtype, especially after subcloning as we did here. Our results therefore directly demonstrate the

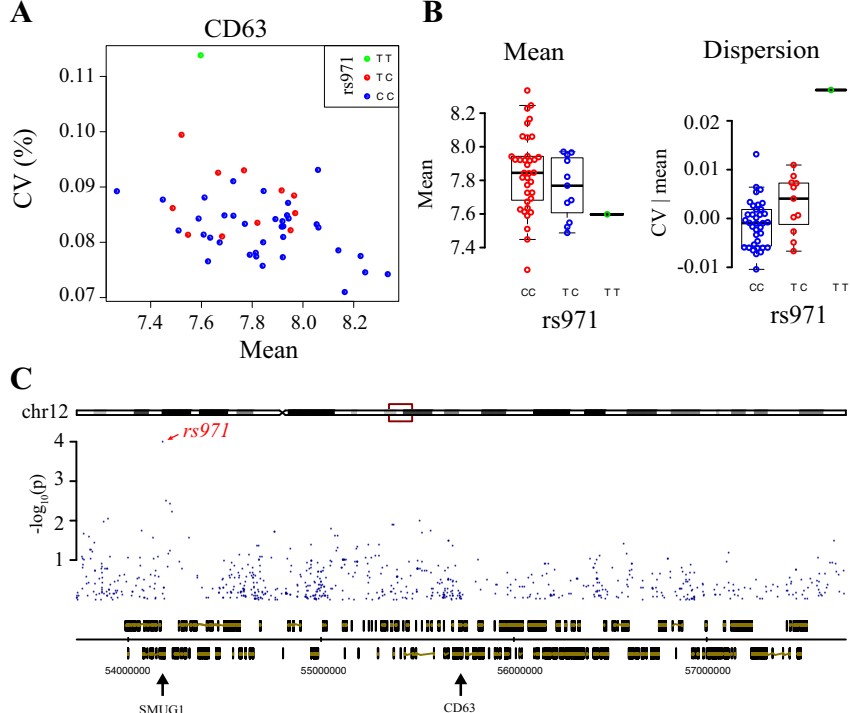

**Fig. 8 Genetic mapping of CD63 expression variability. a** Dot plot of CV versus mean expression of CD63 in 48 LCLs, colored according to their genotype at SNP rs971. **b** Boxplot of CD63 mean expression and dispersion according to *rs971* genotype. Uncorrected linkage *p*-values were 0.1269 for mean expression and 0.0004 for dispersion and corresponded to FDR = 0.998 and 0.1, respectively. **c** Genomic view of the locus. Blue dots, nominal linkage scores for association with CD63 expression variability (CV). Bottom track: genomic coordinates and genes positions. Middle track: transcripts of CD63 and SMUG1. Retrieved from Ensembl on 2019-06-11 using GenomeGraphs[51].

variation of dispersion for a strictly-defined subtype. Secondly, primary cells of different individuals obviously come from different environments (different human bodies). Lu et al. accounted for this by studying the same individuals at consecutive time points, and by using sex and age as covariates, revealing cases where age impacted dispersion. However, various other physiological differences may contribute to dispersion, such as diet, lifestyle, microbiome and the 'exposome'. The effects of all such factors are largely excluded from ex-vivo cultured cell lines because the environment is artificially controlled over a prolonged time. Our results show that variation of dispersion is present despite this controlled context. Finally, unlike in primary cells, gene expression dispersion in LCLs may be affected by EBV infection, for example if EBV changes the amplitude or dynamics of expression in host cells over time. Such effects may shift dispersion levels as compared to non-infected cells and, accordingly, we observed substantial differences in dispersion among LCLs originating from the same donor. However, these differences were weaker than the ones observed between LCLs of unrelated donors. This shows that the possible contribution of EBV on expression dispersion does not dominate the contribution of donor-specific factors, such as genetic variants. The two studies are further complementary by illustrating the contribution of *trans*-acting SNPs in T cells (Lu et al.) and a *cis*-acting SNP in B-derived cells (this work).

Recently, Sarkar et al.[17] searched for *cis*-acting SNPs of expression variability in another cell type: induced pluripotent stem cells, which were derived from the Yoruba LCL lines analyzed here. The authors used single-cell RNAseq, generating data on many more genes but much fewer cells as compared to flow-cytometry. This study failed to identify SNPs affecting dispersion: although SNPs could be associated with variability, this association was fully explained by their effect on mean expression. We

note another layer of complexity in this work: cellular reprogramming may generate variability and it would be interesting to compare various iPSC lines originating from the same LCL. The multiple differences between this study and our present work probably also explain a difference in statistical power: we were fortunate to detect an association with only 50 individuals although Sarkar et al. predicted that, when using single-cell RNAseq data, thousands of individuals would be needed for detection.

We previously described a statistical method (PTLMAPPER) that can use the full distribution of single-cell traits to specifically identify scPTL[11]. This method looks for scPTL that may affect any property of the single-cell expression distribution, and not only specific pre-defined features such as mean, variability or dispersion. We note that when we applied this method to the dataset reported here, we did not detect any statistically-significant association. This is not a discrepancy: we showed earlier that PTLMAPPER and QTL mapping of dispersion traits have different sensitivities and are therefore complementary. So it is not surprising that one of the two methods could detect an association that the other approach missed[11].

What is the biological implication of the genetic modulation of CD23, CD55, CD63, and CD86 expression dispersion? Although the inter-individual differences in dispersion that we report here are subtle they may be important for two reasons. First, a slight probabilistic effect is negligible over small numbers, but becomes clearly substantial over millions of 'trials'. Given the lifetime of a human individual, its high number of B cells and the known activity and reactivity of these cells, the number of times surface proteins are used—the number of 'trials'—is enormous. The second reason is that, for immune cells, what happens to one can matter to all because each cell can trigger a response that is collectively amplified. However, a possible whole-organism

significance of our observations remains to be determined because (i) genetic effects are known to be context-dependent and Petri dishes are radically-different environments than human tissues; (ii) the subtle effects that we described here ex vivo may be hidden by variability resulting from the dynamics of the blood environment (cross-talks with other cell types, variation in nutriments, flow mechanics…); (iii) since molecular and cellular networks can buffer fluctuations, dispersion at the level of one macromolecule does not necessarily imply phenotypic consequences.

The four proteins investigated here have distinct functions and regulators. Expression of CD23, the low-affinity receptor for IgE, is known to be strongly stimulated by EBV-mediated activation[26]. Our observation that CD23 expression is bimodal in EBV-immortalized cell lines from some but not all individuals suggests that this strong activation might be heterogeneous in a genotype-dependent manner. The Decay Accelerating Factor CD55, the tetraspanin CD63 and the CD86 co-stimulator participate to the cross-talk of B cell with T lymphocytes. Thus, different degrees of CD55, CD86, or CD63 expression dispersion in B cells may in turn generate heterogeneities among T cells. In addition, genetic variation in expression dispersion may also have clinical implications. For example, CD55 and CD63 are helpful for the prognosis of various cancers[34–37] but interpretation of these biomarkers may be complicated if their expression dispersion is substantial in some patients and not others despite a similar clinical situation. Furthermore, CD23 is the target of lumiliximab, a monoclonal antibody used to fight chronic lymphocytic leukemia (CLL)[38]. If the patient's genotype modulates CD23 expression bimodality among CLL cells, as we observed here for LCL cell lines, then it probably also modulates the proportion of cells that are efficiently targeted by this treatment.

In conclusion, our observations on a handful of proteins and individuals are important because they demonstrate the existence of non-deterministic genetic effects on gene regulations in humans. It is therefore justified to invest efforts in single-cell considerations when studying the genetic predisposition to certain traits. Such efforts may reveal genetic mechanisms that underly risks of disease emergence or relapse.

## Methods

**Lymphoblastoid cell lines**. Yoruba LCLs were part of the HapMap/1000 Genomes panel and were obtained from the Coriell cell repository (Camden, NJ, USA). Cell lines references are: GM18486A; GM18487; GM18488B; GM18489A, GM18498B; GM18499B; GM18501C; GM18502B; GM18504B; GM18505D; GM18507E; GM18508E; GM18516B; GM18517D; GM18519B; GM18520B; GM18522C; GM18523C; GM18852C; GM18853B; GM18855C; GM18856D; GM18858C; GM18859C; GM18861C; GM18862C; GM18867B; GM18868B; GM18870B; GM18871B; GM18873B; GM18874B; GM18912C; GM18913B; GM18916C; GM18917B; GM18933B; GM18934B; GM19098B; GM19099B; GM19107B; GM19108B; GM19140B; GM19141C; GM19192B; GM19193C; GM19203C; GM19204B; GM19238E; GM19239C. They corresponded to healthy adult donors from 25 YRI families who were unrelated except GM18913B who was the child of GM19238E. Additional LCLs were generated from two individual donors as follows. Primary B cells were freshly isolated from two adult human blood buffy coats by density gradient centrifugation, followed by positive selection using anti-CD19 Immunomagnetic beads (STEMCELL Technologies). Purified B cells (2.10$^5$ cells) were exposed to supernatants of HEK293$_{EBV.B95-8}$ EBV-producing cells[39] at a multiplicity of infection (MOI) of 0.5 infectious particles per B cell, overnight at 37 °C and then washed once with PBS and cultured with RPMI supplemented with 20% FBS until establishment of LCLs (25 days). Twelve LCLs were established from each donor.

**Subcloning of LCL lines**. We isolated monoclonal subclones from five of the LCLs (GM19238E, GM18486A, GM18489A, GM18519B, GM19239C) by counting and diluting cells to a concentration of approximately 1–10 cells/ml. One hundred microliter of this dilution were then added to the wells of 96-well plates that already contained 100 µl of conditioned medium. Conditioned medium was obtained by culturing LCLs to a concentration of 300,000–500,000 cells/ml, pelleting the cells by centrifugation, collecting the supernatant and filtering it through an 0.2 micron Stericup filter (Merck-Millipore). Diluted cells were then left to grow in 200 µl 50%

conditioned media for 3–4 weeks, until growth was observed in some wells, occasionally supplementing the wells with 50% conditioned media to replace evaporated liquid. From the wells where growth was observed, cells were re-plated into 6-well plates and subsequently 25 ml flasks. Finally, vials from each subclone were frozen for further analysis.

**Clonality test by PCR and NGS sequencing of CDR3**. We established a protocol for PCR-based amplification and multiplexed sequencing of the framework 3 VH region based on previous publications[27,28]. To this end, we isolated DNA from cell lines by transferring the cells to 1.5 ml Eppendorf tubes, pelleting them by centrifugation, and lysing them overnight at 65 °C in 500 µl lysis buffer (100 mM Tris pH = 8.5, 5 mM EDTA, 0.2% SDS, 200 mM NaCl), complemented with final concentration of 0.1 mg/ml ProteinaseK. The next day, DNA was precipitated with 350 µl isopropanol at −20 °C for 2 h, centrifuged for 30 min, the supernatant removed, and the pellet washed with 70% ethanol. After additional centrifugation and removal of supernatant the pellet was dried and resuspended in 200 µl TE. We then amplified the region of interest from 1 µl of DNA by a PCR reaction of 35 cycles in a final volume of 15 µl containing an equimolar mixture of primers (2 µM final concentration per primer) extended with Illumina adapters (see primer list in Supplementary Table 2), Platinum Taq (ThermoFisher), and 1.5 mM final MgCl2 concentration. We then performed PCR cleanup by incubating 5 µl of the PCR reaction with 1 µl Exonuclease I (NEB) for 30 min at 37 °C, and inactivating the enzyme by a 15 min incubation at 80 °C. We then used 3 µl of cleaned PCR product as template for a second round of PCR using Illumina Nextera XT Index primers in 50 µl volume. Amplicons quality was checked on agarose gel and we pooled 35 µl from each PCR and performed Exonuclease I cleanup as above. The samples were then split into 4 tubes, run through a Macherey–Nagel PCR cleanup column and each sample was eluted in 30 µl. All products were then pooled, diluted to approx. 150 ng/µl and sequenced by 150 bp paired-end sequencing on a MiSeq Illumina sequencer, generating a total of 12,022,926 pairs of reads (median coverage per sample: 73,198 read pairs). Reads quality was assessed with FastQC (http://www.bioinformatics.babraham.ac.uk/projects/fastqc/, last accessed November 2015). Overlapping paired-end reads were merged using PEAR[40]. We then used bioawk (https://github.com/lh3/bioawk) to extract assembled reads with an average Phred score ≥ 33 and unassembled reads were discarded. FASTQ files were converted into FASTA files using a custom bash function. For mapping, we downloaded Fasta files from http://www.ncbi.nlm.nih.gov/igblast/showGermline.cgi in November 2015 (now migrated at http://www.imgt.org/) and we used standalone IgBLAST version 1.4. with commands C1-C4 of Supplementary Table 3 to create a database of V, D, and J human segments and to assign each read to a specific gene segment, as defined in the IMGT ontology (http://www.imgt.org). Output files were then parsed using a parser of the maintained Galaxy toolshed https://testtoolshed.g2.bx.psu.edu/view/davidvanzessen/igblastparser_igg/176ce910f659 in order to obtain files having one line per read. We then filtered the data by keeping reads meeting all of the following conditions: "CDR3 Found How" ≠ "NOT_FOUND", "VDJ Frame" ≠ "N/A" and "Top D Gene" ≠ "N/A". We excluded singletons (CDR3 sequence covered only once). We removed reads for which CDR3 length was shorter than 5 amino-acids or for which the conserved CDR3/FR4 Phe/Trp-Gly-X-Gly motif in the J region was not present[41]. We considered that a unique CDR3 sequence corresponded to one clone[42]. The percentage of reads matching a given CDR3 sequence was used to define the representativity of the corresponding clone in the sample. When observing the frequencies of the inferred CDR3 sequences in every clone, we saw that, as expected, very few sequences were represented by a large proportion of reads (over ~25%) while numerous sequences had very low frequencies (below ~10%). We therefore chose to consider only the CDR3 sequences that reached 20% of representativity: if multiple sequences reached this threshold, the sample was considered polyclonal; if the major clone reached 80%, the sample was considered monoclonal; otherwise we considered the data to be inconclusive. In Supplementary Table 1, only sequences reaching 20% of representativity are shown. We note that sequence TSGNTGWYSDYWGQG corresponding to GM18505D cell line and subclones was also detected in other samples. In such cases, this incongruity occurred only in one of the two technical replicates; it is therefore not reliably representative of the cell population of the sample. We rather attribute it to a contamination by a DNA amplicon that occurred after cell harvest and DNA extraction.

**LCLs cultivation and immunostaining**. Cells were cultured in Roswell Park Memorial Institute medium (RPMI 1640, GlutaMAX, Thermo Fisher Scientific) supplemented with 15% fetal calf serum (FCS, Eurobio), antibiotics (100 µg/ml penicillin, 100 µg/ml streptomycin). The cells were incubated at 37 °C in a humidified CO$_2$ incubator (5% CO2) in 25 cm$^2$ culture flasks (vented caps). Depending on cell growth, cell lines were diluted two to three times per week. Cell density was maintained between 300,000 cells/ml to less than a million cells/ml in 10 ml final volume of medium to optimize their growth. Although fixation is not mandatory for quantification, we noticed that it reduced day-to-day technical variability and we therefore applied it systematically. For fixation, cells were counted using KOVA slides after a 2-days growth phase (post-passage). Two million cells were collected, adjusted with PBS to a 10 ml volume, followed by centrifugation at 1000 rpm for 5 min at room temperature. Cell pellets were washed with 4 ml PBS, and centrifuged as previously. They were then resuspended in 1.2 ml 4% Paraformaldehyde (PFA)

in PBS and placed on a roller-tube agitator for 20 min at room temperature. Then 4 ml of PBS +2% fetal calf-serum (FCS) were added and cell suspensions were centrifuged at 2000 rpm for 2 min. Pellets were resuspended in 2 ml of PBS + 2% FCS prior to immunolabelling. All labeling steps were performed in 96-well V-bottom plates in which 150 µl of fixed cells (approx. 150,000 cells) were deposited per well. Cells were centrifuged for 2 min at 2000 rpm, and resuspended in 100 µl of PBS + 10% FCS. Subsequently, each well was completed to 200 µl as follows: unlabeled cells (control) were supplemented with 100 µl of PBS + 10% FCS, while for labeling cells we added 100 µl of PBS + 10% FCS containing the antibody at appropriate dilution (See Supplementary Table 4). The plate was then incubated for 1 h at 4 °C followed by a 2000 rpm centrifugation for 2 min. Cells were washed with 200 µl of PBS, followed by another round of centrifugation. Finally, cells were resuspended in 230 µl of freshly diluted DAPI (2.5 µg/ml) in PBS + 1% FCS.

We also performed a specific experiment on a few cell lines to control the possible effect of PFA fixation on CD86 and CD23 immunostaining (Supplementary Fig. 1). In this experiment, cells were split in different aliquots. While some aliquots were processed as above (PFA fixation followed by immunostaining), other aliquots were processed by first immunostaining live cells and then fixing them with PFA prior to acquisitions. For this latter procedure, we resuspended live cells in PBS + 10%FCS, distributed them in V-bottom 96-well plates (150 µl per well, containing ~150,000 cells) that we then centrifuged for 2 min at 2000 rpm. We then added 100 µl of PBS + 10%FCS containing the antibody (or not, for unstained controls) and incubated the plates at 4 °C for 1 h. Cells were then pelleted by centrifugation at 2000 rpm for 2 min, washed with 200 µl of PBS, centrifuged again and pellets were fixed with 200 µl of PBS + PFA 2% for 20 min at room temperature. Cells were then pelleted again, washed with PBS and resuspended in PBS + 1%FCS + 2.5 µg/ml DAPI. For both CD86 and CD23, single-cell expression distributions were strikingly similar when fixation was applied prior to immunostaining or after it (Supplementary Fig. 1).

**Single-cell flow cytometry acquisitions**. Plates were analyzed on a BD FACS-Canto II (BD Biosciences) flow cytometer equipped with a High Throughput Sampler (HTS). Acquisitions parameters were configured as follows: area scaling values of forward scatter (FSC) and side scatter (SSC) channels were optimized with photomultiplier tubes (PMT) voltages at 295 mV and 415 mV, respectively. Fluorescent PMT voltages were adapted for each antibody. DNA content was estimated from DAPI signal acquired in a Pacific Blue filter with PMT set at 280 mV. For each sample, 20,000 cells were acquired.

**Analysis of flow cytometry data: dataset structure**. The study included three sets of experiments that were performed independently and that we treated separately. Set 1 covered the expression levels of 19 proteins (CD2, CD5, CD9, CD19, CD20, CD22, CD23, CD37, CD38, CD40, CD46, CD53, CD55, CD59, CD63, CD79b, CD80, CD86, ROR1) and one isotype control (IgKappa) in 6 LCLs of the 1000Genome project (GM18519B, GM19238E, GM18489A, GM18486A, GM18505D, GM19239C). Set 2 covered the expression levels of 4 proteins only (CD23, CD55, CD63, CD86) in 50 LCLs of the 1000Genome project, in 24 LCLs originating from 2 ungenotyped donors, and also covered single-cell expression levels in subclones derived from Yoruba LCLs of the 1000 Genome Project. Note that for CD63 and CD86, this set contains heterogeneities between the series of 50 LCLs and the series of subclones: some cell lines were analyzed in both series and their mean values were shifted (Supplementary Fig. 3). It would therefore not be appropriate to compare single-cell distributions between the two series. Set 3 corresponded to quantification of CD23 in 6 subclones using a FITC-conjugated antibody (ref 561146 from Pharmingen, Supplementary Table 4) instead of the APC-conjugated anti-CD23 antibody (BD BioSciences, Supplementary Table 4). The following framework was separately applied to each set.

**Analysis of flow cytometry data: data filtering and normalization**. All analysis was done using R (www.r-project.org) and packages flowCore[43] and flowStats[44] from Bioconductor (www.bioconductor.org). For a given experimental data set, data was pre-processed with the following steps. (i) Removal of events with saturated signals. This was done by setting, for each channel, a lower and an upper bound of signal intensity. These boundaries corresponded to the 5th and 95th percentile of all single-event values pooled from all samples. Events falling outside these boundaries in one or more channel were discarded. (ii) For channels where this procedure generated a negative lower bound, all values were augmented by adding to them the absolute value of this lower bound. This avoided negative values in downstream analysis. (iii) Removal of unstained DAPI events. For this, we identified the first motif of peak-valley-peak in the distribution of DAPI intensities by building a distribution density using the stats::density function and passing it to findPeaks() and findValleys() function of the quantmod R package[45]. The estimate of the intensity value of the valley was chosen as a threshold of minimal labeling signal and events below this value were discarded. (iv) Gating on cell size (Supplementary Fig. 4A, B). Cells of homogeneous size were dynamically gated by pooling cells over all samples, defining a perimeter containing 75% of events in the FSC-A, SSC-A plane based on a density kernel and applying this gate to all samples. (v) Removal of doublets (Supplementary Fig. 4C, D). As for cell-size, a gate that contained 80% of remaining events of all samples was then defined in the FSC-

A, FSC-W plane using a density kernel function, and applied to all samples. (vi) Gating of cells in G1 phase of the cell-cycle (Supplementary Fig. 4E, F). For each sample, a threshold of DAPI intensity was set at a value that minimized the first derivative of the density function. Cells with intensity below this threshold were considered to be in the G1 peak of intensity and were kept for further analysis. (vii) Removal of samples with ≤1,000 events. At the end of these steps, about 10,000 cells remained on average per sample. (viii) Correction of fluorescent values for cell size. For each sample, a linear model of the form $y \sim \log(FSC.A) + \log(SSC.A)$ was fitted using the MASS::rlm() function[46] where y was the log intensity of fluorescence. Data of each cell $i$ was then transformed as $y_i = mean(y) + e_i$, where $e_i$ was the $i$-th residual of the model. (ix) Variation between replicates was reduced by grouping all experiments of the same cell line and antibody labeling and applying on this group the flowStats::warpSet() function with default arguments. This function performed re-alignments according to high densities areas. (x) We observed that some samples displayed outlying distributions as compared to their replicated counterparts. To detect these cases in an unbiased way, we estimated all pairwise dissimilarities between replicates (same cell line and antibody staining) using the Kolmogorov–Smirnov statistics computed by the stats::ks.test() function. For each sample belonging to a group of at least three replicates, its dissimilarities to replicate counterparts were averaged and considered as a score $K$ of reproducibility. Samples with $K$ greater than the 95th percentile of all $K$ values were discarded.

**Analysis of flow cytometry data: traits describing cell–cell variability**. Following data pre-processing, cell-to-cell variability within each sample was quantified by the coefficient of variation (CV = sd/mean) of the relevant fluorescent values. To account for sample-to-sample differences in mean expression levels, we also conditioned CV values on mean by computing the residuals of a non-parametric loess regression of CV ~ mean using the stats::loess() function. For CD23 which displayed bimodality, we fitted a 2 components gaussian mixture model (GMM) on expression levels using the Mclust function from package mclust[47] without constraint on parameters. This generated 5 parameters that fully described the distribution observed in each sample: mean and variance of the first component ($\mu_1$ and $\sigma^2_1$), mean and variance of the second component ($\mu_2$ and $\sigma^2_2$), and the proportion of cells (marginal weight) of the first component. For the clustering reported in Fig. 5, we averaged parameter values across replicates to generate five parameters values per LCL. Each parameter was then centered to zero and scaled across the 50 LCLs and we applied hierarchical clustering using complete linkage.

**Genetic linkage: genotypes dataset**. The genotypes of 1000Genome individuals were downloaded from ftp://ftp.1000genomes.ebi.ac.uk/vol1/ftp/release/20130502/ on 13th February 2017. There were 40 individuals where genotyping was at phase 3 (NA19098, NA19099, NA19107, NA19108, NA19141, NA19204, NA19238, NA19239, NA18486, NA18488, NA18489, NA18498, NA18499, NA18501, NA18502, NA18504, NA18505, NA18507, NA18508, NA18516, NA18517, NA18519, NA18520, NA18522, NA18523, NA18853, NA18856, NA18858, NA18861, NA18867, NA18868, NA18870, NA18871, NA18873, NA18874, NA18912, NA18916, NA18917, NA18933, NA18934) and included phased genotypes (one file per chromosome of the hg19 genome release of February 2009, GRCh37 assembly). For 8 additional individuals (NA19140, NA19203, NA18487, NA18852, NA18855, NA18859, NA18862, NA18913), genotypes were unphased and obtained from./supporting/hd_genotype_chip/ in the form of a single file with all chromosomes. Genotypes of 2 individuals were not found on the 1000Genome project server. Annotations of individuals (kinship and sexe) were obtained from file: ftp://ftp.1000genomes.ebi.ac.uk/vol1/ftp/release/20130502/integrated_call_samples_v2.20130502.ALL.ped. We used command lines G1-G4 of Supplementary Table 5 to extract genotypic data corresponding to individuals of our study. We selected variants located on a chromosomic region centered on the transcription start site (TSS) of each gene of interest. positions of these TSS were obtained from http://genome.ucsc.edu/cgi-bin/hgTables downloaded on 22nd February 2017, using the 'txStart' field for genes CD55 and CD86 oriented in the forward direction, and the 'txEnd' field for genes CD23 and CD63 oriented in the reverse direction. Variants located within 2 Mb of the TSS were extracted with command line G5 of Supplementary Table 5. This produced 2 VCF files per gene, which contained the data of either 40 or 48 individuals, in a region of 4 Mb. These were converted to MAP and PED files using command G6 (Supplementary Table 5). Duplicated entries were removed by commands G7-G9 (Supplementary Table 5). Family ID and sex ID (1 for man and 2 for woman) were introduced in the ped file at the 1st and 5th column, respectively. Variants with MAF ≥ 0.05 were kept and variants that failed the Hardy-Weinberg test[48] at a significance threshold of 0.001 were excluded using commands G10–11 (Supplementary Table 5).

**Genetic linkage: association test**. We searched for genetic linkage between expression traits and markers, using either the 48 individuals with unphased genotypes or only the subset of 40 individuals with phased genotypes (see above). Expression traits for CD23, CD55, CD63 and CD86 were mean, CV and CV|mean; and for CD23 we also considered the five traits of the GMM model describing bimodal distributions ($p_1$, $\mu_1$, $\mu_2$, $\sigma_1$, $\sigma_2$). We used PLINK (v1.9)[49] to perform

association tests, using either a purely additive model[50] (command L1 of Supplementary Table 5) or a model that included possible dominance (command L2 of Supplementary Table 5). The model with dominance did not provide additional associations for variability and dispersion traits and all results reported here were obtained with the additive model. Family-wise error rate across all variants was empirically estimated from 10,000 permutations and we retained association where this rate was lower than 0.05 (Table 1). FDR was calculated by using the—adjust option of PLINK (command L1 of Supplementary Table 5). This procedure applied FDR control on each phenotype independently of the linkage results obtained for other phenotypes. The complete results (all SNPs, all traits, with effect size and significance) are available in Supplementary Data 2.

**Reporting summary**. Further information on research design is available in the Nature Research Reporting Summary linked to this article.

## Data availability

The full sequencing data of the clonality assay are available from the European Nucleotide Archive (https://www.ebi.ac.uk/ena/) under study accession number PRJEB37875. Single-cell expression data and SNPs genotypes are available in the BioStudies database (http://www.ebi.ac.uk/biostudies) under accession number S-BSST382. Summaries of expression data, including numbers of cells, as well as GMM parameter values for CD23 are provided in Supplementary Data 1. Complete results of linkage analysis are provided as Supplementary Data 2. Source data of main figures are provided as Supplementary Data 3.

## Code availability

Analysis code is freely available at https://gitbio.ens-lyon.fr/LBMC/gylab/lcl/ under the open source CeCill licence.

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

## Acknowledgements

We thank Mirko Francesconi and Olivier Gandrillon for critical reading of the manuscript, Thierry Defrance, Evelyne Manet, Jacqueline Marvel for discussions, Abhishek Sarkar and two anonymous referees for peer reviewing, Laura Presti and Celia Joseph for technical contribution, Véronique Barateau and SFR Biosciences Gerland-Lyon Sud (UMS344/US8) for access to flow cytometers and technical assistance, the 1000 Genome Project Consortium for access to cell lines and genotypes, the Pôle Scientifique de Modélisation Numérique for computing resources, BioSyL Federation and Ecofect Labex for inspiring scientific events, Laurent Modolo for guidelines and tools related to code versioning, developers of IgBLAST, PLINK, R, git, bioconductor, and Ubuntu for their software. This work was supported by the European Research Council under the European Union's Seventh Framework Programme FP7/2007–2013 Grant Agreement no. 281359 (G.Y.) and EMBO long-term fellowship ALTF 691-2014 (O.S.).

## Author contributions

Performed experiments: G.T., O.S., H.G. Developed analysis tools: C.B., S.J., G.Y. Analyzed the data and interpreted results: G.T., O.S., C.B., G.Y. Conceived and supervised the study: G.Y. Designed the study: O.S. and G.Y. Wrote the paper: G.Y.

## Competing interests

The authors declare no competing interests.
