## [Peer Review File · Communications Biology]

Reviewers' comments:

Reviewer #1 (Remarks to the Author):

The paper presents evidence suggesting that variation in gene expression variability (variation in protein abundance between cells within an individual) can be partially explained by genetic variation between individuals (QTLs), even after accounting for the fact that variation in mean gene expression could also explain variation in gene expression variability. This work builds upon previous studies, which either did not find QTLs which acted on variability independently of acting on the mean (Sarkar et al. 2019), or did not eliminate environmental effects on variability (Lu et al. 2016).

The experimental approaches used appear to be sound. However, I am worried that weaknesses of the statistical analysis compromise the main result of the paper.

Major points:

1. The p-values reported in Table 1 are reported to be corrected for the number of SNPs corrected, but not what error rate was being controlled. The methods report FWER was controlled, which needs to be reported in the table legend and main text.

I think it is a serious flaw that the p-values were not also corrected for the number of phenotypes tested. It is also not clear from the methods whether the association results were corrected for the fact that each SNP-phenotype combination was tested twice, once with an additive model and once with a dominance model.

It appears the single dispersion QTL reported in the paper would not be significantly associated at the stated FWER if the p-values were also corrected for these sources of multiplicity. This problem appears to seriously compromise the main result of the paper.

This dispersion QTL may still be significant after controlling the FDR (e.g. Benjamini-Hochberg procedure) instead. It is the case that controlling FDR over each phenotype independently controls FDR over the entire study.

Related, the complete results of association analysis (effect size, standard error, and p-value for all SNPs tested, regardless of significance) need to be made available.

2. To account for bimodal gene expression of CD23, the paper reports results of fitting a two-component GMM to the data.

However, the paper reports not all cell lines displayed bimodal gene expression. I found it very difficult to read this result off of the hierarchical clustering plotted in Fig 5.

The defining feature of clusters 2 and 3 appears to be that the two Gaussians have $\mu_1 = \mu_2$, and $\sigma_1^2 \neq \sigma_2^2$ (i.e., they are described by a scale mixture of Gaussian). This suggests a simpler analysis, where you perform a model comparison (e.g., comparing BIC reported by Mclust) of the two models:

M1: 2-component GMM with varying means and varying scales
M2: 2-component GMM with common mean and varying scales

This analysis would directly label gene expression as bimodal or not. I think the main result is robust (some lines display unimodal gene expression, and some display bimodal gene expression), but could be more clearly justified by this more straightforward analysis.

It would also be helpful to plot the observed data (histogram) super-imposed with the fitted mixture distribution in the examples (Fig 5C), to demonstrate the goodness of fit.

3. The paper suggests, and seeks to eliminate, two effects of EBV which could invalidate the conclusions of the paper: (1) EBV could alter the expression dispersion; (2) multiple subclones were immortalized by EBV. To eliminate (1), the paper reports an experiment showing that variation within lines derived from a single donor is less than the variation between lines of different donors.

It appears that de novo lines derived from donor 2 sometimes have mean gene expression (e.g. CD63) and expression dispersion (e.g. CD55) outside of the range observed for the 50 YRI cell lines (Fig 6). But presumably at least one line from donor 2 is represented among those original 50 cell lines? It would be helpful to understand: (1) why the de novo lines appear to be so different from the previously established lines, (2) whether these differences suggest that EBV could **systematically** change gene expression dispersion, (3) what are the implications of (2) on the interpretation of dispersion QTLs which could be discovered using LCLs.

Minor points:

1. The section on covariance of expression dispersion is interesting, but seems to be a digression from the main message of the paper. I think the paper could be more focused and easier to read by removing it.

2. I am confused by the conceptual framework established in the introduction. An scPTL is defined to be a variant which has a non-deterministic effect on cell-level phenotypes. The idea is that scPTLs change the distribution of the phenotype across a population of cells, and the specific case of interest in this study is scPTL which control gene expression variability.

However, as I understand it, established theory suggests that **deterministic** effects can alter the steady state distribution of mRNA in a population of cells (Munsky et al. Science 2013; Kim & Marioni Genome Biology 2013). This theory has motivated others to study QTLs which (deterministically) alter kinetic parameters of transcription (e.g. Larsson et al. Nature 2019), and through those effects change the steady state distribution of mRNA in a population of cells.

It would be helpful to clarify the connection between the scPTL concepts and these previous ideas and results.

3. I think several claims need to be reworded more precisely:

- title: "dispersion...displays genetic variation". I think it would be more

precise to say "dispersion is linked to genetic variation".

- l. 147: "variation in CV could not directly attributed to variation of the mean". I think it would be more precise to say "not completely explained by variation in the mean".

- l. 227: "significantly caused by inter-individual variation". I would suggest only using "significantly" to describe the results of a statistical test. I think it would be more precise to say "primarily driven by inter-individual variation"

- l. 367 "did not explain the differences". I think it be more precise to say "did not completely explain".

4. Some essential descriptive statistics of the experiments are missing or conflicting:

- Fig 1 suggests the Set 1 data has 10,000 cells per line per replicate (after gating). It would be helpful to report the exact numbers in a supplementary table/figure.

The analogous numbers for Set 2 and Set 3 are not given.

- The main text reports 18 proteins were selected for the Set 1 data; the methods report 20 proteins.

7. I was surprised that no association for mean gene expression was found; however, Battle et al. Science 2015 also found no associations for the five proteins tested in this study. It would be helpful to cite this previous result.

5. Fig 2A: the error bars are very difficult to read, and in many cases are obscured by the symbols denoting the mean values across the replicates.

6. Fig 8 (and Fig S1): it would be helpful to show boxplots of phenotype stratified by genotype at the QTL for mean also, to show no effect on mean gene expression.

--

Dr. Abhishek K. Sarkar
Department of Human Genetics, University of Chicago

Reviewer #2 (Remarks to the Author):

The manuscript "Cell-to-cell expression dispersion of B-cell surface proteins displays genetic variation among humans" from Gérard Triqueneaux, Claire Burny, Orsolya Symmons and co-authors is an interesting work. They make use of established B-cell lines derived from a previously genotyped Yoruba population. They combine the use of genetic analysis and flow cytometry to infer "cell-cell expression variation" of known B-cell markers and underlying association to specific genotypes. The manuscript is well written and scientifically sound. However, I have some questions related to technical aspects of the paper:

1) The authors use PFA to fix the cells prior to staining. Fixation is known to have an effect on epitope conformation and the binding of several clones is heavily affected by it

(<https://www.biolegend.com/newsdetail/5122/>), which can have effects not only on mean but especially on signal variation. I would suggest the inclusion of controls comparing the displayed measures under non-fixed conditions, to improve the strength of the paper conclusions.

2) Figure 7 is an essential figure of the paper to exclude that the differences in variance observed arent a mere measure of cell clonality. However, the authors display only part of the data generated and to strength their point I would suggest that the data is displayed for every clone generated (in 7C and 7D).

3) Still related to figure 7, I think it is problematic the inclusion of clone 8E-8E4. Based on the CDR3 region sequencing, it is highly likely this clone was contaminated with 5D cells prior to library generation, as the CD23 expression pattern looks like a hybrid of 8E plus 5D. This leads me to doubt the robustness of the other data and at least that clone should be excluded of the manuscript and no conclusions should be supported by such clone.

4) How did the authors establish 20% as a threshold to defining true clones identified in the CDR3 sequencing?

Reviewer #3 (Remarks to the Author):

General Comments

The authors have presented a fairly extensive investigation of potential explanations for the observations of cell-to-cell expression variability for 5 particular cell surface proteins in cultured human lymphoblastoid cell lines. These include effects such as within-individual variation, clonality and a limited examination of genetic influences.

Overall the authors should be commended for addressing these issues and going to the extent of demonstrating the reproducibility of dispersion within an individual, as well as the potential for EBV-transformation and LCL clonality to explain expression heterogeneity. The former should be emphasised more, i.e. there is a very high consistency for dispersion estimates from the same individual. However, it seems that the authors fall short in their explanations for these phenomena which could be investigated with relatively straightforward experiments. In particular, the observation that bimodal expression of CD23 emerges in sub-clones, as well as its presence in the LCLs of several Yoruban individuals. Testing for active EBV expression, say by RT-qPCR for instance, might provide a simple experimental demonstration of this.

Furthermore, I have concerns about the robustness of the genetic analyses given the relatively small sample size, and recent report by Sarkar et al. that $n \gg 4000$ is required to detect cis-acting dispersion QTLs. In particular, more analytical details and a demonstration of the robustness and replicability of their genetic association would convince the reader that these findings are not a false-positive.

More generally, whilst the authors have gone to great pains to try to explain their observed expression heterogeneity, it is hard to discern what the exact hypothesis is in the first place. I feel that contextualising their observations in the form of an initial hypothesis would help the reader to interpret the potential importance and impact of these results. This subsequently leads to a very weak overall conclusion, which leaves the reader somewhat unconvinced.

Specific comments

In addition to the above, I have specific technical and compositional comments, broken down into each section of the manuscript.

Abstract:

Line 5: "...but whether such effects exist in humans has not been fully demonstrated." - strictly speaking Lu et al. and Wills et al. (both cited) were the first to demonstrate genetic influences on

expression noise – this statement is therefore false.

Line 13: "Such subtle genetic effects may participate to phenotypic variation and disease predisposition" – This seems like a pretty big stretch at the moment – as far as I'm aware now has yet demonstrated that expression noise contributes to disease.

Introduction:

Overall the introduction would benefit from a description of the known sources of expression heterogeneity, rather than in the discussion, and framing of the hypothesis before summarising the findings.

Line 23: "Quantitative genetics is challenged by..." – this is rather strange phraseology as quantitative genetics provides the explanation for complex trait inheritance. I'm not sure what point the authors are trying to make here.

Line 39: "It is possible that such loci contribute to disease..." – again, pretty wild speculation – there is no direct (or circumstantial) evidence for this.

Line 62: "...strict probabilistic effects" – the appearance of a probabilistic effect does not necessarily make it stochastic. For instance, an ensemble of conflicting deterministic effects can give the appearance of stochasticity, such as consensus voting as is observed in the choice between lytic and lysogenic stages in lambda phage. This limitation should be recognised in the manuscript, rather than making an appeal to randomness as an explanation.

Results:

Lines 94-114: Quantifying expression dispersion using millions of lymphoblastoid cells – this section seems more suited to the introduction, whilst a justification for the selection of proteins to study (perhaps in table format?) would be highly beneficial to the reader. Particularly with respect to the biological relevance of these proteins under study.

Lines 138-145: It is not clear why one protein would be selected for further study over another. The choices seem arbitrary. A more robust or systematic, i.e. quantitative, approach for studying these proteins should be provided. For instance, is there more variability in expression than would be expected by chance?

Line 147: "...variation in CV could not be directly attributed to variation of mean." – I think there should be a formal comparison here, otherwise this statement is unsupported.

Likewise on line 148: "...was significantly lower..." – by what measure or comparison was it significantly lower? Was this formally tested?

Lines 155-158: A justification or motivation for studying expression dispersion of the target proteins should be provided. This would greatly aid the reader in being able to critically appraise the authors study, and discern its relevance.

Line 158: Where is CD19? – it was selected for further study based on lines 145-146.

Lines 180-188: It is not clear what the purpose of this analysis is, nor the hypothesis for why it is performed, let alone an indication of its biological relevance or importance.

Lines 268-278: These sections are very methodological or include technical details that are not of specific interest and would be better represented in 1) the methods, and 2) in tabular format for read numbers, etc.

Lines 314-319: What is the significance of this lack of differences in clonality, and can it be

formally tested? Comparing plots is arbitrary and something of a “dark art”, rather than rigorous science.

Genetic mapping of expression variability and dispersion. This section lacks important analytical details. For instance, how was the analysis performed, how many variants – was genetic relatedness accounted for? What are the effect sizes, and in what direction, i.e. increase or decrease dispersion? These are crucial details for interpreting the genetic association results and should be included as a minimum standard. What was the a priori threshold for significance, and did this account for multiple testing burden?

Overall, this section lacks replication and robustness (with the exception of the extra genotyping of the CD23-associated variant). Additionally, how is the reader supposed to interpret the possible association between SMUG1 and CD63 – are these genes related in any way? Is this association altering expression of SMUG1 or other genes in the vicinity? There is a wealth of data that could be leveraged to properly address this role of genetic variation and expression dispersion.

Discussion:

Line 365: “found” should be replaced with “confirmed” – we already know that expression dispersion differs between individuals.

Line 372: “Consistently, we found a cis-acting SNP...” – this implies a degree of replication and robustness that has not been demonstrated in this manuscript.

Lines 404-406: “The effects of all such factors are excluded...” – strictly speaking any environmental exposure can impact on a phenotype up until the point at which the cells were sampled for immortalisation.

Lines 418-425: The authors should address the biggest message from the Sarkar et al. paper, and that is their sample size calculations recommend $n \gg 4000$ – how do the authors address this clear barrier in their own study?

Lines 427-434: If the scPTL approach did not yield any findings, a) why mention it at all, b) what is the reason for the discrepancy between these different analyses?

Lines 436-437: Only one genetic association is reported for CD63 – why are CD23, CD55 and CD86 mentioned in this context?

Line 450: “...phenotypic alterations.” – I think “phenotypic consequences” might be more accurate.

Lines 451 – 469: This is the justification that should be used for the selection of the proteins, i.e. it should be in the introduction or methods!

Lines 471-473: I found these conclusions quite weak and empty – why are these results important? What is their actual relevance, not wild speculation.

Methods:

Line 549 – why is the 20% threshold used? Can this be justified?

Line 625 – “..the derivative...” – the first derivative, the second, which one?

Figures:

Throughout the figures the error bars are unreadable – these should use a darker colour and thicker line width to make them obvious.

Figure 1 – this relates to the text – but what is the observed deviation from the mean-variance

relationship greater than expected by chance? The sampling distribution of the CV is undefined, by could the authors determine an empirical null distribution by permutation?

Figure 2 – What do the asterisks denote? These are hard to read and the same information could be portrayed by highlighting the relevant panels for the proteins that were selected for further study. This would be much easier to read.

Figure 4 – The pie charts are entirely redundant – why not just state the correlation on the scatter plots, with the actual p-value rather than asterisks. What does the loess line add to the figure? Furthermore, how robust are the co-dispersions between proteins?

Figure 7 – The arrangement of panel C is confusing. Maybe have 1 row or column per parental line. This also raises the question – are the sub-clones more similar to their parental line than they are to other (sub) clones?

Figure 8 – Panels B and C are unreadable. The medians in panel B are obscured and unreadable, whilst panels A and B show redundant information.

Point-by-point response to reviewers

We thank all three reviewers for their comments which helped us improve the manuscript. Please find below a detailed response to all the points that have been raised. The relevant text revisions are highlighted in yellow in the provided files.

Reviewer #1 (Remarks to the Author):

The paper presents evidence suggesting that variation in gene expression variability (variation in protein abundance between cells within an individual) can be partially explained by genetic variation between individuals (QTLs), even after accounting for the fact that variation in mean gene expression could also explain variation in gene expression variability. This work builds upon previous studies, which either did not find QTLs which acted on variability independently of acting on the mean (Sarkar et al. 2019), or did not eliminate environmental effects on variability (Lu et al. 2016).

The experimental approaches used appear to be sound. However, I am worried that weaknesses of the statistical analysis compromise the main result of the paper.

Major points:

1. The p-values reported in Table 1 are reported to be corrected for the number of SNPs corrected, but not what error rate was being controlled. The methods report FWER was controlled, which needs to be reported in the table legend and main text.

I think it is a serious flaw that the p-values were not also corrected for the number of phenotypes tested. It is also not clear from the methods whether the association results were corrected for the fact that each SNP-phenotype combination was tested twice, once with an additive model and once with a dominance model. It appears the single dispersion QTL reported in the paper would not be significantly associated at the stated FWER if the p-values were also corrected for these sources of multiplicity. This problem appears to seriously compromise the main result of the paper. This dispersion QTL may still be significant after controlling the FDR (e.g. Benjamini-Hochberg procedure) instead. It is the case that controlling FDR over each phenotype independently controls FDR over the entire study.

We agree with the reviewer and we have now computed the FDR (Benjamini-Hochberg procedure, implemented in PLINK) for every linkage scan. We are happy to report that the dispersion QTL for CD63 still remains significant (FDR = 0.006). This conclusion as well as FDR values have been added to the manuscript (Table 1 and methods).

Related, the complete results of association analysis (effect size, standard error, and p-value for all SNPs tested, regardless of significance) need to be made available.

All linkage results, regardless of significance, are now provided as Supplementary Data S2. These include the outcome of all SNP-trait tests: effect size, standard error, p-values and FDR. Please see the included README.md file for details.

2. To account for bimodal gene expression of CD23, the paper reports results of fitting a two-component GMM to the data.

However, the paper reports not all cell lines displayed bimodal gene expression. I found it very difficult to read this result off of the hierarchical clustering plotted in Fig 5.

The defining feature of clusters 2 and 3 appears to be that the two Gaussians have $\mu_1 = \mu_2$, and $\sigma_1^2 \neq \sigma_2^2$ (i.e., they are described by a scale mixture of Gaussian). This suggests a simpler analysis, where you perform a model comparison (e.g., comparing BIC reported by Mclust) of the two models:

M1: 2-component GMM with varying means and varying scales

M2: 2-component GMM with common mean and varying scales

This analysis would directly label gene expression as bimodal or not. I think the main result is robust (some lines display unimodal gene expression, and some display bimodal gene expression), but could be more clearly justified by this more straightforward analysis.

The color matrix of the initial version of Fig. 5B corresponded to scaled parameter values: each column was scaled and centered independently. As a result, colors could only be compared within a given column and not across columns. We understand that this representation was confusing because it seemed that, for many rows, σ_1 and σ_2 values were markedly different and that the sign of this difference changed between the two clusters. In fact, variances of components are not the only cause of the separation between clusters 2 and 3: for example, μ_1 values are largely different between the two clusters (visible on the initial figure). This is also true for μ_2 but to a lower extent. To avoid this confusion, and since the important information of Fig. 5B was the clustering tree, we removed the color matrix from the revised figure and we now provide the GMM parameter values in Supplementary Data S1. We also indicate the IDs of the cell lines on the clustering tree in Supplementary Fig. S4.

Please note that we use GMM parameters to describe distributions and compare them across cell lines, and not to test them individually for bimodality. In case readers want to assess the relevance of the mixtures, they can now consult the BIC values reported in Supplementary Data S1.

It would also be helpful to plot the observed data (histogram) super-imposed with the fitted mixture distribution in the examples (Fig 5C), to demonstrate the goodness of fit.

The revised figure (Fig. 5B) now shows the model components super-imposed with the observed data. We removed the initial Fig. 5A panel to avoid redundancy.

3. The paper suggests, and seeks to eliminate, two effects of EBV which could invalidate the conclusions of the paper: (1) EBV could alter the expression dispersion; (2) multiple subclones were immortalized by EBV. To eliminate (1), the paper reports an experiment showing that variation within lines derived from a single donor is less than the variation between lines of different donors.

*It appears that de novo lines derived from donor 2 sometimes have mean gene expression (e.g. CD63) and expression dispersion (e.g. CD55) outside of the range observed for the 50 YRI cell lines (Fig 6). But presumably at least one line from donor 2 is represented among those original 50 cell lines? It would be helpful to understand: (1) why the de novo lines appear to be so different from the previously established lines, (2) whether these differences suggest that EBV could *systematically* change gene expression dispersion, (3) what are the implications of (2) on the interpretation of dispersion QTLs which could be discovered using LCLs.*

There are many differences between the Yoruban LCLs and the *de novo* LCLs from donors 1 and 2. First, for obvious confidentiality reasons, the origin of donors 1 and 2 was not passed to us but the samples were likely collected in France or nearby shortly before LCL establishment. It is therefore extremely unlikely that the ethnical origin of these donors was related to Yorubans. Second, we also don't know the physiological status (pathologies?) of these donors at the time of collection. Third, the

EBV infection was done in a different laboratory and therefore likely under different conditions regarding media, growth times, EBV particles etc... Fourth, Yoruba cell lines had been maintained by the Coriell Institute who sent us frozen aliquots, whereas donors 1 and 2 lines were analyzed directly on site after their amplification.

Given all these differences, our design is suitable to compare donors 1 and 2 within themselves and from one another, but not to compare them to the Yoruba set. One could even be surprised to see that the data is not further different between these two unrelated sets of experiments.

Minor points:

1. The section on covariance of expression dispersion is interesting, but seems to be a digression from the main message of the paper. I think the paper could be more focused and easier to read by removing it.

The extent of covariation is a question we frequently have, if not always, from the audience when presenting at conferences. We therefore think it makes sense to report it in the main text. We have fused this section with the previous one to preserve the main flow.

2. I am confused by the conceptual framework established in the introduction. An scPTL is defined to be a variant which has a non-deterministic effect on cell-level phenotypes. The idea is that scPTLs change the distribution of the phenotype across a population of cells, and the specific case of interest in this study is scPTL which control gene expression variability.

*However, as I understand it, established theory suggests that *deterministic* effects can alter the steady state distribution of mRNA in a population of cells (Munsky et al. Science 2013; Kim & Marioni Genome Biology 2013). This theory has motivated others to study QTLs which (deterministically) alter kinetic parameters of transcription (e.g. Larsson et al. Nature 2019), and through those effects change the steady state distribution of mRNA in a population of cells.*

It would be helpful to clarify the connection between the scPTL concepts and these previous ideas and results.

We have clarified this by adding a sentence in introduction: "Note that, if the trait results from a stochastic process, a deterministic effect on a kinetic parameter of the process can have inherently probabilistic effects at the cell level. For example, a change of transcriptional burst frequency and size may modify the probability that one cell expresses a gene at a given level at any given time without necessarily affecting mean expression (Kim and Marioni 2013)".

3. I think several claims need to be reworded more precisely:

- title: "dispersion...displays genetic variation". I think it would be more precise to say "dispersion is linked to genetic variation".

The revised title now reads "...dispersion is linked to genetic variants...".

- l. 147: "variation in CV could not directly attributed to variation of the mean". I think it would be more precise to say "not completely explained by variation in the mean".

We changed this as suggested.

- l. 227: "significantly caused by inter-individual variation". I would suggest only using "significantly" to describe the results of a statistical test. I think it would be more precise to say "primarily driven by inter-individual variation"

Yes, thank you. We changed this as suggested.

- I. 367 "did not explain the differences". I think it be more precise to say "did not completely explain".

Yes, we changed it by "did not fully explain".

4. Some essential descriptive statistics of the experiments are missing or conflicting:

- Fig 1 suggests the Set 1 data has 10,000 cells per line per replicate (after gating). It would be helpful to report the exact numbers in a supplementary table/figure. The analogous numbers for Set 2 and Set 3 are not given.

For data Set 1, the median number of cells after gating is 5740 so we modified Fig1 which now reads "~6000 cells per sample". The exact number of cells prior and after gating are now provided in Supplementary Data S1 for every sample (for Set 1, Set 2 and Set 3). We also added in these tables the values of mean expression and cv.

- The main text reports 18 proteins were selected for the Set 1 data; the methods report 20 proteins.

We initially screened 19 proteins and we also included an isotype control (IgKappa), making a total of 20 immunostainings. One protein (ROR1) gave signals that were not beyond background. The isotype control gave very weak signal (as expected). This is now explained in the revised main text and methods.

7. I was surprised that no association for mean gene expression was found; however, Battle et al. Science 2015 also found no associations for the five proteins tested in this study. It would be helpful to cite this previous result.

Thank you for checking this! Indeed, this study identified 2 eQTLs (for mRNA) but no pQTLs for the 4 genes covered here. We have added this comment in the revised text (lines 362-365).

5. Fig 2A: the error bars are very difficult to read, and in many cases are obscured by the symbols denoting the mean values across the replicates.

We revised the figure and removed the symbols: we now plot only the error bars so that dots correspond to mean +/- sem. We did this for Figs 2A, 3A, 6A and 7D. We also darkened the error bars of Fig. 1B.

6. Fig 8 (and Fig S1): it would be helpful to show boxplots of phenotype stratified by genotype at the QTL for mean also, to show no effect on mean gene expression.

We have added a boxplot of CD63 mean values stratified by genotype in revised Fig 8. We did not modify Fig S1 because the large spread of mean values within all genotypic categories is apparent from the dot plots.

--

Dr. Abhishek K. Sarkar
Department of Human Genetics, University of Chicago

Reviewer #2 (Remarks to the Author):

The manuscript "Cell-to-cell expression dispersion of B-cell surface proteins displays genetic variation among

humans" from Gérard Triqueneaux, Claire Burny, Orsolya Symmons and co-authors is an interesting work. They make use of established B-cell lines derived from a previously genotyped Yoruba population. They combine the use of genetic analysis and flow cytometry to infer "cell-cell expression variation" of known B-cell markers and underlying association to specific genotypes. The manuscript is well written and scientifically sound. However, I have some questions related to technical aspects of the paper:

1) The authors use PFA to fix the cells prior to staining. Fixation is known to have an effect on epitope conformation and the binding of several clones is heavily affected by it (<https://www.biolegend.com/newsdetail/5122/>), which can have effects not only on mean but especially on signal variation. I would suggest the inclusion of controls comparing the displayed measures under non-fixed conditions, to improve the strength of the paper conclusions.

We thank the reviewer for this important comment. We performed an additional control experiment for CD23 and CD86, the two proteins for which expression dispersion is particularly different between cell lines. We processed cells in parallel along two protocols: one corresponding to our initial procedure of PFA fixation followed by immunolabelling, and the other protocol corresponding to immunolabelling of live cells followed by PFA fixation. We now show in Supplementary Fig. S1 that the two protocols generate strikingly similar distributions. In particular, cell lines that we initially described to have high CD86 dispersion or bimodal CD23 distribution clearly display these features under non-fixed conditions. This conclusion was added to the main text and methods, together with the details of this specific experiment.

2) Figure 7 is an essential figure of the paper to exclude that the differences in variance observed are not a mere measure of cell clonality. However, the authors display only part of the data generated and to strengthen their point I would suggest that the data is displayed for every clone generated (in 7C and 7D).

For the initial submission, we only produced a partial characterization of the subclones, because of limited resources. Our goal was to obtain sufficient examples that would rule out the attribution of high dispersion to polyclonality. The data we showed were the entire data we had produced. After the reviewer's comment, we tried to thaw and re-amplify some of the clones but this proved difficult because many cultures grew very poorly.

3) Still related to figure 7, I think it is problematic the inclusion of clone 8E-8E4. Based on the CDR3 region sequencing, it is highly likely this clone was contaminated with 5D cells prior to library generation, as the CD23 expression pattern looks like a hybrid of 8E plus 5D. This leads me to doubt the robustness of the other data and at least that clone should be excluded of the manuscript and no conclusions should be supported by such clone.

We understand this concern. There are two reasons why we believe that the contamination was not at the level of cells but at the level of DNA during CDR3 genotyping (most likely an amplicon from 5D hitting other DNA samples). First, the contaminating sequence of 5D is present not only in the sequencing data from this clone but also in the data from 3 other subclones of the same 8E line as well as in the data from 2 subclones of the 9B line. Given our culturing conditions, we don't see how cells could have contaminated 6 different cultures. Second, the IgPCR and NGS sequencing was done twice on every clone (technical duplicates of amplification and seq); so if the contamination originated from cells, duplicates should both detect it. We note that for clone 8E-8E4 the 5D-related sequence was detected in only one of the replicates and clearly not in the other one. This is also the case for clones 9B-2F10 and 9B-2B5 where the contaminating sequence was detected in only one of the duplicates. We therefore think that a PCR product from the amplification of the CDR3 region of 5D contaminated

some of the other DNA samples prior to sequencing. We have added these details and arguments in the revised methods (lines 591-596)

4) How did the authors establish 20% as a threshold to defining true clones identified in the CDR3 sequencing?

No sample showed 100% of reads assigned to a single CDR3 sequence. This is most likely due to technical low-rate errors during PCR, sequencing, reads-mapping and/or locus reconstruction. When we observed the frequencies of the inferred sequences in every clone, we saw that very few sequences were represented by a large proportion of reads (over ~25%) while numerous sequences had very low frequencies (below ~10%). As examples, we show here the frequencies for the clone discussed above.

8E-8E4 replicate 1:

8E-8E4 replicate 2:

Choosing 20% as a cutoff was just a simple way to distinguish these two types of sequences. This is now explained in the methods section (lines 584-588).

Reviewer #3 (Remarks to the Author):

General Comments

The authors have presented a fairly extensive investigation of potential explanations for the observations of cell-to-cell expression variability for 5 particular cell surface proteins in cultured human lymphoblastoid cell lines. These include effects such as within-individual variation, clonality and a limited examination of genetic influences.

Overall the authors should be commended for addressing these issues and going to the extent of demonstrating the reproducibility of dispersion within an individual, as well as the potential for EBV transformation and LCL clonality to explain expression heterogeneity. The former should be emphasised more, i.e. there is a very high consistency for dispersion estimates from the same individual.

Thank you. We have added a sentence making this emphasis (line 268).

However, it seems that the authors fall short in their explanations for these phenomena which could be investigated with relatively straightforward experiments. In particular, the observation that bimodal expression of CD23 emerges in sub-clones, as well as its presence in the LCLs of several Yoruban individuals. Testing for active EBV expression, say by RT-qPCR for instance, might provide a simple experimental demonstration of this.

It is true that EBV expression may change between cells from a same lineage and this, in turn, could affect CD23 expression dispersion. We plan to further clarify this point in a later study that will be focused on CD23: by measuring EBV expression at the single-cell level (as suggested by the reviewer) and on pools of cells sorted on the basis of CD23 expression. We also plan to study CD23 expression bimodality on primary cells and its possible implication on lumiliximab immunotherapy. Observations reported in the current manuscript constitute a basis for this follow-up characterization.

Furthermore, I have concerns about the robustness of the genetic analyses given the relatively small sample size, and recent report by Sarkar et al. that $n \gg 4000$ is required to detect cis-acting dispersion QTLs. In particular, more analytical details and a demonstration of the robustness and replicability of their genetic association would convince the reader that these findings are not a false-positive.

We now provide more details on the association results. In particular, the FDR is now provided, which also supports significance for the results presented in Table 1, and all results are available as Supplementary Data S2. The contrast between our small sample size and the previous power estimation is now mentioned in the revised text (lines 455-459). For a complete replicability study, one would need to re-profile numerous additional lines, and we cannot currently afford this additional work. We therefore have toned down our claims on rs971:

- Abstract: we changed "We linked" by "We detected a genetic association";
- Introduction: we changed "we identified a cis-acting SNP that affects..." by " we detected a significant association between the expression dispersion of CD63 and the genotype at a SNP located in cis ";
- Results, we added: " Note that our observations do not fully demonstrate the effect of rs971 on CD63 dispersion because i) the genetic association needs to be replicated using another sample of individuals and ii) the mechanism..."
- Discussion now also clearly mentions the need for replicability (lines 401-403).

More generally, whilst the authors have gone to great pains to try to explain their observed expression heterogeneity, it is hard to discern what the exact hypothesis is in the first place. I feel that contextualising their observations in the form of an initial hypothesis would help the reader to interpret the potential

importance and impact of these results. This subsequently leads to a very weak overall conclusion, which leaves the reader somewhat unconvinced.

We revised the introduction which now enters the study directly and more clearly via the angle of cell-cell heterogeneity and emphasizes on the importance to assess scPTLs in humans. We also revised the last sentences of the discussion so that they echo as an answer to the important question of the existence of non-deterministic effects.

Specific comments

In addition to the above, I have specific technical and compositional comments, broken down into each section of the manuscript.

Abstract:

Line 5: "...but whether such effects exist in humans has not been fully demonstrated." - strictly speaking Lu et al. and Wills et al. (both cited) were the first to demonstrate genetic influences on expression noise – this statement is therefore false.

We changed this by "...but very few studies have detected such effects in humans."

Line 13: "Such subtle genetic effects may participate to phenotypic variation and disease predisposition" – This seems like a pretty big stretch at the moment – as far as I'm aware now has yet demonstrated that expression noise contributes to disease.

The link between expression noise and disease is now particularly well established for bacterial persistence to antibiotics (eg. work of Balaban et al.) as well as anti-cancer drug resistance: Baskar 2019 doi: 10.26508/lisa.201900554, Iliopoulos 2011 doi: 10.1073/pnas.1018898108, Sharma 2010 doi: 10.1016/j.cell.2010.02.027, Shaffer 2017 doi: 10.1038/nature22794. A genetic effect in this context may not cause a "predisposition" *per se* but rather affect the personal response to treatment. We have therefore changed "predisposition" by "outcome".

Introduction:

Overall the introduction would benefit from a description of the known sources of expression heterogeneity, rather than in the discussion, and framing of the hypothesis before summarising the findings.

As mentioned above, introduction now directly mentions cell-cell heterogeneity (rather than incomplete penetrance and small-effects genetic variants). The sources of heterogeneity have already been discussed extensively so we now cite helpful reviews.

Line 23: "Quantitative genetics is challenged by..." – this is rather strange phraseology as quantitative genetics provides the explanation for complex trait inheritance. I'm not sure what point the authors are trying to make here.

This part is no longer present in the revised introduction.

Line 39: "It is possible that such loci contribute to disease..." – again, pretty wild speculation – there is no direct (or circumstantial) evidence for this.

Yes, this is largely speculative so far. We re-phrased by "...scPTL may contribute to disease predisposition" and we cite an earlier paper where we previously discussed this speculation, especially regarding autosomal dominant diseases (TIGS 2014). We now also cite a very helpful and more recent review highlighting the link between noise and drug escapers (Jolly *Frontiers Oncol* 2018).

Line 62: "...strict probabilistic effects" – the appearance of a probabilistic effect does not necessarily make it stochastic. For instance, an ensemble of conflicting deterministic effects can give the appearance of stochasticity, such as consensus voting as is observed in the choice between lytic and lysogenic stages in lambda phage. This limitation should be recognised in the manuscript, rather than making an appeal to randomness as an explanation.

We removed this ambiguous assertion at the end of the sentence.

Results:

Lines 94-114: Quantifying expression dispersion using millions of lymphoblastoid cells – this section seems more suited to the introduction, whilst a justification for the selection of proteins to study (perhaps in table format?) would be highly beneficial to the reader. Particularly with respect to the biological relevance of these proteins under study.

We understand this comment but we prefer to walk the reader through this again. Regarding the biological relevance of the proteins: it now appears much earlier (shortly after these lines) for the four proteins that we really characterized.

Lines 138-145: It is not clear why one protein would be selected for further study over another. The choices seem arbitrary. A more robust or systematic, i.e. quantitative, approach for studying these proteins should be provided. For instance, is there more variability in expression than would be expected by chance?

Our focus was inter-individual differences in variability. Quantifying variability *per se*, especially as compared to expectation from chance only, would require another design. To make this clearer, we introduce this text by "... and we examined if one or more of the proteins displayed different CV across individuals."

Line 147: "...variation in CV could not be directly attributed to variation of mean." – I think there should be a formal comparison here, otherwise this statement is unsupported.

As suggested by reviewer 1, we changed this assertion by "...was not completely explained by...". The examples cited in the following sentences provide illustrations of why this is the case. We also added a Kruskal-Wallis test on Fig 3C demonstrating that this is the case.

Likewise on line 148: "...was significantly lower..." – by what measure or comparison was it significantly lower? Was this formally tested?

A *t*-test comparing the CD23 CV of two cell lines with similar means is indicated on the revised Fig 2A and legend. For CD55, this test was not significant with only 6 lines, so we revised the text by "...seemed to have a larger CV" and we provide the formal Kruskal-Wallis test with all 50 LCLs on Fig. 3C.

Lines 155-158: A justification or motivation for studying expression dispersion of the target proteins should be provided. This would greatly aid the reader in being able to critically appraise the authors study, and discern its relevance.

As suggested by the reviewer, the biology of these proteins is now introduced here rather than in the discussion.

Line 158: Where is CD19? – it was selected for further study based on lines 145-146.

In fact we decided to reliably characterize only four of the five proteins on the full set of 50 LCLs. This was mainly due to a technical problem that delayed our analysis of CD19 on a set of LCLs.

Lines 180-188: It is not clear what the purpose of this analysis is, nor the hypothesis for why it is performed, let alone an indication of its biological relevance or importance.

The revised text now introduces this analysis with the question of a global vs. modular variation, and concludes on modularity.

Lines 268-278: These sections are very methodological or include technical details that are not of specific interest and would be better represented in 1) the methods, and 2) in tabular format for read numbers, etc.

We moved the two read numbers to methods.

Lines 314-319: What is the significance of this lack of differences in clonality, and can it be formally tested? Comparing plots is arbitrary and something of a "dark art", rather than rigorous science.

The revised text now provides *t*-tests results on both CV and mean for the examples initially mentioned (all were significant). These examples are now pointed by arrows on the revised Fig. 7D.

Genetic mapping of expression variability and dispersion. This section lacks important analytical details. For instance, how was the analysis performed, how many variants – was genetic relatedness accounted for? What are the effect sizes, and in what direction, i.e. increase or decrease dispersion? These are crucial details for interpreting the genetic association results and should be included as a minimum standard. What was the a priori threshold for significance, and did this account for multiple testing burden?

As mentioned above and in response to Reviewer 1, analytical details are now provided in the revised manuscript, including FDR control for multiple testing, effect sizes and direction. The complete association results are provided as Supplementary Data.

Overall, this section lacks replication and robustness (with the exception of the extra genotyping of the CD23-associated variant). Additionally, how is the reader supposed to interpret the possible association between SMUG1 and CD63 – are these genes related in any way? Is this association altering expression of SMUG1 or other genes in the vicinity? There is a wealth of data that could be leveraged to properly address this role of genetic variation and expression dispersion.

We agree that replicability is needed for a complete demonstration and we now explicitly mention it in text. We also toned down our claims on *rs971*. Regarding possible mechanism: we did not see anything sufficiently sound to be mentioned. We agree that it is tempting to build hypothesis on annotations (such as hitting a regulatory element, having a cis-eQTL effect etc...) but, from our experience in yeast - where data is even more wealthy - we learned that when the hypothesis is vague, it is usually also wrong. Especially when the regulatory SNP resides in a gene of unrelated function (such as SMUG1 here).

Discussion:

Line 365: "found" should be replaced with "confirmed" – we already know that expression dispersion differs between individuals.

We revised the sentence. To be more precise on the novelty with respect to Lu *et al.*, we added "... even within a single cell subtype and under controlled conditions".

Line 372: "Consistently, we found a cis-acting SNP..." – this implies a degree of replication and robustness that has not been demonstrated in this manuscript.

Yes. The revised text now explicitly mentions the need for replicability in the next sentence.

Lines 404-406: "The effects of all such factors are excluded..." – strictly speaking any environmental exposure can impact on a phenotype up until the point at which the cells were sampled for immortalisation.

We wrote "...are largely excluded..." in the revised text.

Lines 418-425: The authors should address the biggest message from the Sarkar et al. paper, and that is their sample size calculations recommend $n \gg 4000$ – how do the authors address this clear barrier in their own study?

There are multiple differences between the two studies: RNAseq vs. flow-cytometry, cell types, designs... The data structure is therefore different and we think it is hard to transpose the power calculation from Sarkar *et al.* to the present context. These considerations are now mentioned (lines 455-459).

Lines 427-434: If the scPTL approach did not yield any findings, a) why mention it at all,

Because negative results are results, and some readers are likely interested in this methodological consideration.

b) what is the reason for the discrepancy between these different analyses?

There is no discrepancy. We showed in the scPTL paper (Chuffart 2016) that the two methods are complementary because they have different sensibilities. We revised the text to make this point more explicit.

Lines 436-437: Only one genetic association is reported for CD63 – why are CD23, CD55 and CD86 mentioned in this context?

This section relates to dispersion as a genetic trait, not to the variants causing it. It is true that we did not detect association for CD23, CD55 and CD86, but our observations support that genetic variation exists. We are therefore in the - now common - situation of being able to observe genetic variation but unable to find the genetic determinants, most likely because of statistical power.

Line 450: "...phenotypic alterations." – I think "phenotypic consequences" might be more accurate.

We changed as suggested.

Lines 451 – 469: This is the justification that should be used for the selection of the proteins, i.e. it should be in the introduction or methods!

We moved this part upstream at the beginning of the results section.

Lines 471-473: I found these conclusions quite weak and empty – why are these results important? What is their actual relevance, not wild speculation.

The revised text now explicitly explains why our study is important: the examples that we describe justify future efforts (lines 506-508).

Methods:

Line 549 – why is the 20% threshold used? Can this be justified?

Please see our response to reviewer 2 point 4 above.

Line 625 – “..the derivative...” – the first derivative, the second, which one?

We now indicate it is the first derivative.

Figures:

Throughout the figures the error bars are unreadable – these should use a darker colour and thicker line width to make them obvious.

We revised figures 2A, 3A, 6A and 7D and removed the symbols: we now plot only the error bars so that dots correspond to mean +/- sem. We also darkened the error bars of Fig. 1B.

Figure 1 – this relates to the text – but what is the observed deviation from the mean-variance relationship greater than expected by chance? The sampling distribution of the CV is undefined, by could the authors determine an empirical null distribution by permutation?

As explained above, we now cover these considerations on Fig 2A (for CD23) and 3C (for all four proteins).

Figure 2 – What do the asterisks denote? These are hard to read and the same information could be portrayed by highlighting the relevant panels for the proteins that were selected for further study. This would be much easier to read.

Thank you for this suggestion, we followed it.

Figure 4 – The pie charts are entirely redundant – why not just state the correlation on the scatter plots, with the actual p-value rather than asterisks. What does the loess line add to the figure? Furthermore, how robust are the co-dispersions between proteins?

We removed the pie charts and loess lines, and we now show correlation values and actual p-values on the scatter plots.

Figure 7 – The arrangement of panel C is confusing. Maybe have 1 row or column per parental line. This also raises the question – are the sub-clones more similar to their parental line than they are to other (sub) clones?

We now present this panel with one column per parental line.

Figure 8 – Panels B and C are unreadable. The medians in panel B are obscured and unreadable, whilst panels A and B show redundant information.

It is true that Panel A alone provides the full information. We nonetheless followed the recommendation of Reviewer 1 and we provide a more complete panel B including a boxplot of dispersion vs. genotype. Also, medians are now thicker on B and we removed the RNA isoforms from panel C to make it more readable.

REVIEWERS' COMMENTS:

Reviewer #1 (Remarks to the Author):

The revised manuscript addresses many of my previous concerns. I have some additional minor points which should be addressed:

1. I previously suggested applying the Benjamini-Hochberg procedure to control the FDR of reported associations at some fixed level (e.g., FDR 10% as in Sarkar et al. 2019). The revised manuscript reports the output values of this procedure generated by PLINK in Table 1 as FDR.

However, those values are more precisely described as "corrected p-values" than false discovery rates. Taking corrected p-values < 0.05 , say, is equivalent to controlling the FDR at level 0.05.

Estimating the FDR for a specific test (i.e., posterior probability of a false discovery) requires a more sophisticated procedure, e.g. locfdr (Efron 2005) or ashR (Stephens 2016).

I also suggest:

- report the FDR at which tests are reported to be significant (or not significant) in the main text

- clarify in the methods section whether the FDR procedure was applied independently to each of the phenotypes analyzed

2. I previously made suggestions to clarify Fig 5. The changes have improved the figure, but I have some additional suggestions:

- It might be clearer to plot the unsmoothed data histograms, in order to distinguish between the data and the fitted models

- If the three panels in Fig 5B are taken from specific individuals, it would be helpful to mark them in Fig 5A to clarify the connection

3. I previously asked why the distribution of protein expression in the two additional donors was so different from the YRI individuals first analyzed.

I suggest clarifying in the main text that the additional donors were not Thousand Genomes individuals.

Reviewer #3 (Remarks to the Author):

The authors have done an excellent job at addressing many of the concerns that I raised with their original submission. I have one remaining issue with the genetic association section, namely that I am not wholly convinced that their association between rs971 and CD63 variability (CV) does not depend on the mean. For this effect to be divorced from (even subtle) changes in mean expression I believe they should present the equivalent analysis for the dispersion (CV conditioned on mean), that they present elsewhere in the manuscript.

Point-by-point response to reviewers

We thank the reviewers for their latest comments.

Reviewer #1 (Remarks to the Author):

The revised manuscript addresses many of my previous concerns. I have some additional minor points which should be addressed:

1. I previously suggested applying the Benjamini-Hochberg procedure to control the FDR of reported associations at some fixed level (e.g., FDR 10% as in Sarkar et al. 2019). The revised manuscript reports the output values of this procedure generated by PLINK in Table 1 as FDR. However, those values are more precisely described as "corrected p-values" than false discovery rates. Taking corrected p-values < 0.05, say, is equivalent to controlling the FDR at level 0.05. Estimating the FDR for a specific test (i.e., posterior probability of a false discovery) requires a more sophisticated procedure, e.g. locfdr (Efron 2005) or ashr (Stephens 2016).

Yes we agree. In this sense, the values we computed are similar to q -values that are often used in genomics studies. By computing these values with PLINK, we could revise the manuscript without changing the entire linkage test procedure. We hope this point is now clearer with the latest revisions of text and methods (see below).

I also suggest:

- report the FDR at which tests are reported to be significant (or not significant) in the main text

Although we computed the FDR, we kept our initial criteria for significance (FWER < 5%). We added the following sentence to the main text: "We considered an association to be significant if its family-wise error rate was lower than 5% and we estimated, for each trait, the False Discovery Rate (FDR) corresponding to the retained associations."

- clarify in the methods section whether the FDR procedure was applied independently to each of the phenotypes analyzed

Yes it was and we explicitly wrote it in the revised methods (line 764).

2. I previously made suggestions to clarify Fig 5. The changes have improved the figure, but I have some additional suggestions:

- It might be clearer to plot the unsmoothed data histograms, in order to distinguish between the data and the fitted models

The revised figure now shows histograms of the data instead of smoothed densities.

- If the three panels in Fig 5B are taken from specific individuals, it would be helpful to mark them in Fig 5A to clarify the connection

We added labels (stars) in Fig. 5A.

3. I previously asked why the distribution of protein expression in the two additional donors was so different from the YRI individuals first analyzed.
I suggest clarifying in the main text that the additional donors were not Thousand Genomes individuals.

We added this statement in the results section (line 252).

Reviewer #3 (Remarks to the Author):

The authors have done an excellent job at addressing many of the concerns that I raised with their original submission. I have one remaining issue with the genetic association section, namely that I am not wholly convinced that their association between rs971 and CD63 variability (CV) does not depend on the mean. For this effect to be divorced from (even subtle) changes in mean expression I believe they should present the equivalent analysis for the dispersion (CV conditioned on mean), that they present elsewhere in the manuscript.

The association result for dispersion (CV conditioned on mean) is shown in Fig. 8B (right). For clarity, we added in its legend the corresponding p-value (nominal = 0.0004, corresponding to FDR = 0.1). It did not pass the multiple-testing threshold when scanning all SNPs (and therefore did not appear in Table 1) but the fact that it is small shows that correction for the mean did not erase the association. Consistently, there is no association with the mean, as shown in Fig. 8B (left).